# Revisiting Anisotropy in Language Transformers: The Geometry of Learning Dynamics

Raphael Bernas [1]    Fanny Jourdan [2 3]    Antonin Poché [2 4]    Céline Hudelot [1]

## Abstract

Since their introduction, Transformer architectures have dominated Natural Language Processing (NLP). However, recent research has highlighted an inherent anisotropy phenomenon in these models, presenting a significant challenge to their geometric interpretation. Previous theoretical studies on this phenomenon are rarely grounded in the underlying representation geometry. In this paper, we extend them by deriving geometric arguments for how frequency-biased sampling attenuates curvature visibility and why training preferentially amplify tangent directions. Empirically, we then use concept-based mechanistic interpretability during training, rather than only post hoc, to fit activation-derived low-rank tangent proxies and test them against ordinary backpropagated true gradients. Across encoder-style and decoder-style language models, we find that these activation-derived directions capture both unusually large gradient energy and a substantially larger share of gradient anisotropy than matched-rank normal controls, providing strong empirical support for a tangent-aligned account of anisotropy.

## 1. Introduction

Representations produced by Transformer language models (Vaswani et al., 2017) are often *anisotropic*: instead of spreading roughly uniformly, token (or sentence) embeddings concentrate in a narrow cone and exhibit high pairwise cosine similarity even for unrelated content (Mu & Viswanath, 2018; Ethayarajh, 2019; Gao et al., 2019; Biś et al., 2021). This geometric bias was initially framed as a

pathology because it degrades similarity-based reasoning, complicates probing and linear separation, and can distort semantic geometry for downstream uses such as retrieval or sentence embedding pipelines (Timkey & van Schijndel, 2021; Mu & Viswanath, 2018). Empirically, anisotropy has been linked to training and modeling details, including likelihood maximization with weight tying and the induced "representation degeneration" of embeddings (Gao et al., 2019; Zhang et al., 2020), as well as the influence of rare-token gradients and frequency effects (Yu et al., 2022; Zhou et al., 2021; Diehl Martinez et al., 2024a).

However, recent work suggests that anisotropy should not be treated as a purely negative artifact. Some conclusions in the literature depend on brittle isotropy metrics (Rudman et al., 2022); fully isotropic spaces are not always compatible with clustered representations and linear classification objectives (Mickus et al., 2024); and explicit regularization of embedding geometry can even favor retaining or increasing anisotropy during training (Rudman & Eickhoff, 2024). Moreover, recent work suggests that anisotropy may be inherent to the self-attention mechanism itself, rather than solely a consequence of training objectives (Godey et al., 2024). Additionally, anisotropy is coherent with concentration along a few dominant directions, effectively reducing the *intrinsic* dimensionality of the representation space. This observation is consistent with the fact that successful fine-tuning and generalization often operate in low-dimensional subspaces despite massive overparameterization (Aghajanyan et al., 2021; Razzhigaev et al., 2024). From this perspective, anisotropy may reflect an implicit dimension reduction that supports generalization in overparameterized models.

A recurring concern, however, is that anisotropy may also be partly explained by the syntactic nature of language data and by how syntax is geometrically encoded in Transformer representations. Prior work shows that dependency structure can be embedded in representation space and that syntax can occupy prominent subspaces (Hewitt & Manning, 2019; Coenen et al., 2019; Yokoi et al., 2024). If dominant geometric factors are driven by syntactic regularities, this may limit semantic disentanglement and make both model behavior and mechanistic analysis harder to interpret: humans

[1]MICS, CentraleSupelec, Université Paris-Saclay [2]IRT Saint Exupéry, Toulouse, France [3]Mila - Quebec AI Institute, Montreal, Canada [4]IRIT Toulouse, Toulouse, France. Correspondence to: Raphael Bernas <raphael.bernas@centralesupelec.fr>.

*Proceedings of the 43rd International Conference on Machine Learning*, Seoul, South Korea. PMLR 306, 2026. Copyright 2026 by the author(s).

may then "see" geometry that largely reflects syntax and frequency rather than meaning (Zhou et al., 2021). Despite the close connection between anisotropy and latent-space geometry, anisotropy has rarely been studied through an explicitly *geometric* lens that accounts for local structure and on-manifold phenomena. Recent evidence suggests that token spaces may be better described as stratified objects with measurable curvature, rather than smooth manifolds (Robinson et al., 2026; 2025). Building on this viewpoint, we make the following contributions:

- We provide theoretical arguments that the underlying *syntactic geometry* of language favors a low-dimensional linear organization, and thus may favor anisotropic representations.

- We connect these geometric biases to Transformer learning dynamics, highlighting architectural and training mechanisms that preferentially amplify such directions.

- We empirically test the theory across encoder-style and decoder-style checkpoints by extracting activation-derived concept subspaces during training and showing that true-gradient energy and anisotropy are preferentially aligned with these tangent proxies; we further report supplementary mechanistic-interpretability (Olah et al., 2020) analyses of representation geometry during learning.

## 2. Theoretical Arguments

### 2.1. Manifold hypothesis for language embeddings

Let $W_E \in \mathbb{R}^{V \times d}$ be a learned token embedding matrix (with $V$ the vocabulary size and $d$ the hidden dimension). Let $\mathcal{T}_\theta : \mathbb{R}^d \to \mathbb{R}^d$ denote a transformer block and let $\mathcal{P}_\omega$ denote a prediction head, respectively parameterized by $\theta$ and $\omega$. For a model with $L$ blocks we write

$$f_{\theta,\omega}(w) = \mathcal{P}_\omega \circ \mathcal{T}_\theta^{(L)} \circ \cdots \circ \mathcal{T}_\theta^{(1)}\left(W_E^\top e_w\right),$$

with $e_w$ the one-hot for token $w$. The classical manifold hypothesis asserts that high-dimensional observations concentrate near a differentiable submanifold $\mathcal{M} \subset \mathbb{R}^d$ of low intrinsic dimension; this hypothesis motivates manifold learning and manifold-aware algorithm (Roweis & Saul, 2000; Belkin et al., 2006; Fefferman et al., 2016).

For language embeddings we adopt a *local manifold* (or stratified-manifold) view: in sufficiently small semantic neighborhoods the empirical distribution of embedding vectors concentrates near a low-dimensional differentiable manifold or a small union of manifolds $\bigcup_k \mathcal{M}_k \subset \mathbb{R}^d$. This is deliberately weaker than a global manifold claim and suffices for local tangent-space approximations, intrinsic-dimension estimates and geometric arguments used later in

this paper. Recent empirical and theoretical work complicates a naive global manifold assumption: Robinson et al. (2026) demonstrate that token subspaces often fail manifold (and even smooth fiber-bundle) models, exhibiting statistically detectable local structure that is incompatible with a single smooth manifold hypothesis. Complementary work shows that the token subspace topology can be recovered up to homeomorphism, which supports viewing token geometry through a topological or stratified lens rather than as a single global smooth manifold (Robinson et al., 2025).

Accordingly, for the remainder of this paper we assume the *local manifold hypothesis*: in every sufficiently small semantic neighborhood there exists a $C^3$ manifold that approximates the data. This framework allows us to treat token embeddings not as static points, but as samples from distributions defined over these local manifolds. In the following section, we characterize a dominant bias in this regime: the frequency-dependent shrinkage of the sampling variance as measured in the ambient Euclidean metric.

### 2.2. Empirical Check: Frequency-Induced Variance Collapse & the Embeddings Sampling Law

Prior research has established a negative dependence between token frequency and embedding norm, demonstrating that frequent tokens exhibit significantly reduced magnitudes compared to rare tokens (Oyama et al., 2023; Liang et al., 2021). We posit that this norm shrinkage has not only a geometric effect, but constrains the *sampling law* of the token representations within the local syntactic manifold $\mathcal{M}$.

Formally, let $\mu_i$ denote the "idealized" centroid of the embedding for token $i$. We define the deviation of a sampled realization $v_i$ from this centroid as $t = \|v_i - \mu_i\|$. Let $g_i(t)$ be the probability density function governing this deviation. Our hypothesis is that for high-frequency tokens, the shrinkage of the global norm forces $g_i(t)$ to concentrate sharply around low values of $t$. In contrast, rare tokens retain a more diffuse distribution, allowing for greater geometric expressivity. This implies that frequent tokens are effectively "pinned" to their idealized form $\mu_i$, limiting the variance available to the model when sampling these tokens during contextual processing.

To empirically characterize $g_i(t)$, we leverage the stochastic nature of the training process. While the inference model is deterministic, the training phase subjects the model to continuous stochastic updates (via mini-batch selection and optimizer dynamics). We can thus view the sequence of embeddings across training checkpoints as discrete realizations drawn from the model's internal sampling distribution. Let $W_E^{(k)} \in \mathbb{R}^{V \times d}$ represent the embedding matrix at checkpoint step $k$. We approximate the sampling law for token $i$

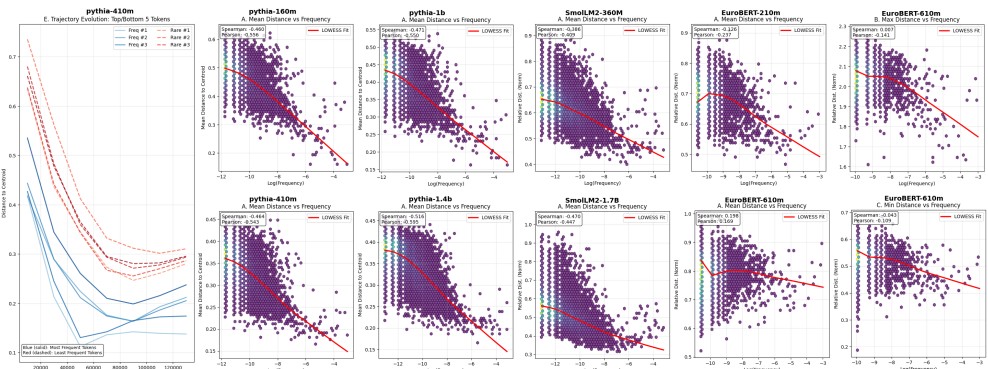

*Figure 1.* **Temporal dynamics of embedding variance across model architectures. Left:** Training trajectories for the top-5 most frequent versus bottom-5 least frequent tokens in PYTHIA-410m, illustrating the tighter confinement of frequent tokens. **Middle:** Mean Euclidean distance to centroid vs. log-frequency for PYTHIA (160m, 410m, 1b, 1.4b), SMOLLM2 (360m, 1.7b), and EUROBERT (210m, 610m). Causal decoders exhibit a strong negative correlation (Pearson $\rho \approx -0.5$), confirming frequency-induced variance collapse. **Rightmost:** Detailed Min/Max distance statistics for EUROBERT-610m. Although the mean trend is attenuated, the distributional view still shows a weaker frequency-dependent concentration pattern.

using the set of realizations $\{v_i^{(k)}\}_{k=k_{start}}^{K}$.

Crucially, we set $k_{start}$ to a post-warmup phase rather than initialization. Input embeddings, particularly for frequent tokens, are the first parameters to structurally stabilize due to their high update frequency during backpropagation. By ignoring the initial transient phase ($k < k_{start}$), we ensure that the observed variation $\{v_i^{(k)}\}$ reflects the inherent *sampling noise* (the "shaking" of the converged concept) rather than the trajectory of learning from random initialization.

We check our hypothesis across the PYTHIA (Biderman et al., 2023), SMOLLM (allal et al., 2025), and EUROBERT (Boizard et al., 2025) suites. Figure 1 quantifies the relationship between token frequency and the geometric stability of embeddings during training. For the causal decoder architectures (PYTHIA and SMOLLM2 suites), we observe a robust negative correlation (Pearson $\rho \approx -0.5$) between log-frequency and the mean distance to the centroid. This empirically validates that frequent tokens are subjected to a highly concentrated sampling law $g_i(t)$, forcing them to converge rapidly and oscillate within a restricted volume of the ambient space. In contrast, rare tokens maintain larger trajectory variances, retaining higher kinematic freedom throughout the optimization process.

The EUROBERT models exhibit a weaker version of this trend, with substantially attenuated mean correlations ($\rho \approx -0.2$ for the 210m variant and $\rho \approx 0.1$ for the 610m variant). We therefore do not treat the mean statistic alone as decisive in this case. However, the rightmost panel of Figure 1 shows that the distributional picture is not flat: for EUROBERT-610m, the minimum and maximum deviation statistics still move mildly but consistently in the expected

direction, indicating that frequency bias continues to leave a trace even if the aggregate effect is attenuated. In that sense, EUROBERT is better viewed as a weaker instance of the same concentration phenomenon than as a contradiction to it. We provide some intuition on a plausible explanation: First, the Masked Language Modeling (MLM) objective, utilizing a masking rate (0.5), may impose different gradient dynamics than the autoregressive likelihood maximization of decoders. Second, EUROBERT explicitly optimizes for multilingual parity through data sampling strategies designed to ensure equivalent representation across languages (Boizard et al., 2025). This balancing likely counteracts the natural over-representation bias of the corpus, preventing high-frequency tokens from dominating the optimization landscape to the same extent observed in standard English-dominant decoders. We empirically confirmed that for frequent tokens, the approximated density $g_i(t)$ is condensed, whereas for rare tokens, it allows farther spread.

## 2.3. Bias on Manifold

We work in the embedding space $\mathbb{R}^d$ produced by a standard encoder (the learned embedding matrix $W_E$ or the output of a transformer). We adopt the Data Manifold Hypothesis in a local sense and assume there exists an idealized semantic manifold $\mathcal{M} \subset \mathbb{R}^d$ of class $C^3$ which would capture the canonical semantic configurations from which word vectors are (approximately) sampled. Importantly, we do *not* require global manifoldity: our theoretical claims require only that, locally, the distribution concentrates near a $C^3$ submanifold (or a finite union / a stratified family of such submanifolds). This local assumption suffices for our tangent-space expansions and curvature bounds; if a true semantic manifold

exists globally and is $C^3$, identical statements follow. Our goal in this section is to (i) state a minimal set of geometric and sampling assumptions that formalize the intuition developed above, (ii) record the extrinsic second-order expansion used to relate intrinsic displacements to ambient Euclidean distances, and (iii) show how a radial sampling bias induced by standard embedding procedures systematically reduces the empirical visibility of high-curvature directions.

**Notation and standing assumptions.** Let $\mu \in \mathcal{M}$ be a reference point (the semantic center). For $x \in \mathcal{M}$ close to $\mu$ write $v \in T_\mu\mathcal{M}$ for the tangent component so that $x$ corresponds to an intrinsic displacement $v$ from $\mu$. Denote $r = \|v\|$ and let $u = v/r$ be the tangent direction. We assume:

(A1) ($C^3$ manifold + reach) $\mathcal{M}$ is a compact $C^3$ embedded Riemannian submanifold of $\mathbb{R}^d$ with positive reach $\tau > 0$. In particular the nearest-point projection is well-defined and unique in a tubular neighbourhood of radius $\tau$ (Smolyanov et al., 2007).

(A2) (Local extrinsic regularity) the embedding $\iota : \mathcal{M} \hookrightarrow \mathbb{R}^d$ has uniformly bounded second fundamental form II and bounded third derivatives on the neighbourhood of $\mu$ we consider (this controls the $C^3$ remainder constants in local expansions) (Lang, 1999).

(A3) (Radial sampling model) samples are drawn according to an ambient density $p_{\mathrm{amb}}(y) = g(t)$ with $t = \|y - \mu\|$, where $y$ is the sampled embedding associated to the idealized $\mu$ and $g$ is strictly decreasing when $t$ increase. This models the empirical relation $t = \phi(f)$, high-frequency words concentrate at small $t$ (Oyama et al., 2023; Liang et al., 2021).

**Arc-chord comparison and curvature.** For clarity we distinguish intrinsic (arc) and ambient (chord) distances. Let $r := d_\mathcal{M}(\mu, y)$ be the intrinsic (geodesic) distance and $t := \|y - \mu\|$ the ambient Euclidean chord. Define the directional curvature scalar $\kappa(u) := \|\mathrm{II}_\mu(u, u)\|^2$. The following classical expansion (valid for small $r$) isolates curvature:

$$t^2 = r^2 - \tfrac{1}{12}\,\kappa(u)\,r^4 + \mathcal{O}(r^5). \tag{1}$$

(For formula (1) see Proposition 6 or Lemma 16 in (Smolyanov et al., 2007; Gao, 2021)).

**Sampling bias induced by ambient-distance dependence.** Fix a small ambient chord $t > 0$ and consider tangent directions $u$ for points at that ambient distance from $\mu$. The following proposition formalizes the *compressive* effect of curvature on the relationship between intrinsic and ambient distances.

**Proposition 2.1** (Compressing effect of curvature). *Under (A1)-(A3), fix a small ambient radius $t > 0$. For two unit tangent directions $u, u'$, the intrinsic distances satisfy, to leading order,*

$$r^2(t, u) - r^2(t, u') = \tfrac{1}{12}\left(\kappa(u) - \kappa(u')\right)t^4 + \mathcal{O}(t^5).$$

*At fixed ambient radius $t$, directions with larger $\kappa(u)$ correspond to larger intrinsic distances $r$.*

*Proof.* See Appendix A □

**Density-theoretical formulation.** We now derive the probabilistic consequences of Proposition 2.1 under the ambient sampling model (A3). Let $k = \dim \mathcal{M}$. In intrinsic polar coordinates around $\mu$, the Riemannian volume element includes an intrinsic curvature correction (Lang, 1999): $d\nu = \left(1 + \mathcal{R}(u)\frac{r^2}{6} + \mathcal{O}(r^3)\right)r^{k-1}\,dr\,d\sigma(u)$, where $d\sigma$ is the uniform measure on $S^{k-1} \subset T_\mu\mathcal{M}$ and $\mathcal{R}(u) := \mathrm{Ric}(u, u)$ is the Ricci curvature in direction $u$. This accounts for how the intrinsic geometry of $\mathcal{M}$ distorts volume, distinct from the extrinsic curvature $\kappa(u)$ captured by the second fundamental form. Points are sampled with ambient density $g(t)$, so the joint density on $\mathcal{M}$ in coordinates $(t, u)$ is

$$p(t, u) \propto g(t) \cdot \left(1 + \mathcal{R}(u)\frac{r^2}{6}\right) \cdot r(t, u)^{k-1} \cdot h(u) \cdot \left|\frac{\partial r}{\partial t}\right|, \tag{2}$$

where $h(u)$ is the intrinsic angular prior and the Jacobian $|\partial r/\partial t|$ arises from the coordinate change.

**Corollary 2.2** (Curvature-induced directional bias). *Under (A1)-(A2), at fixed small ambient radius $t$, the log-ratio of conditional probabilities for two directions $u, u'$ satisfies*

$$\log \frac{p(u \mid t)}{p(u' \mid t)} = \log \frac{h(u)}{h(u')} + \left(\tfrac{1}{6}(\mathcal{R}(u) - \mathcal{R}(u'))\right.$$
$$\left. + \tfrac{k+2}{24}(\kappa(u) - \kappa(u'))\right)t^2 + \mathcal{O}(t^3).$$

*If $h$ is approximately uniform, directions are biased by both intrinsic curvature $\mathcal{R}(u)$ (from volume distortion) and extrinsic curvature $\kappa(u)$ (from embedding compression).*

*Proof.* See Appendix B. □

This curvature bias is a purely geometric effect arising from (A1)-(A2): at any fixed ambient shell, the combined influence of intrinsic volume distortion (Ricci curvature $\mathcal{R}$) and extrinsic embedding geometry (second fundamental form $\kappa$) determines which directions are over-represented. Crucially, this bias does not depend on the radial sampling law $g(t)$ and would, in principle, enhance the embedding expressivity by making curved regions more visible.

**Attenuation of curvature bias under radial concentration.** While $g(t)$ cancels in the conditional $p(u \mid t)$, it

critically shapes the *marginal* distribution of directions. The marginal density on $u$ is

$$p(u) = \int_0^T p(u \mid t)\, p(t)\, dt \propto h(u)\Big[ \int_0^T g(t)\, t^{k-1} dt + \Big(\mathcal{R}(u)\tfrac{1}{6} + \tfrac{k+2}{24}\kappa(u)\Big) \int_0^T g(t)\, t^{k+1} dt \Big], \tag{3}$$

where $T$ is the validity range. Define the moment ratio

$$\eta_g := \frac{\int_0^T g(t)\, t^{k+1}\, dt}{\int_0^T g(t)\, t^{k-1}\, dt}.$$

Then

$$\begin{aligned}
p(u) &\propto h(u)\Big(1 + \Big(\mathcal{R}(u)\tfrac{1}{6} + \tfrac{k+2}{24}\kappa(u)\Big)\eta_g\Big) \\
&= h(u)\Big(1 + \tfrac{1}{24}\big(4\mathcal{R}(u) + (k+2)\kappa(u)\big)\eta_g\Big).
\end{aligned} \tag{4}$$

Since $g(t)$ is decreasing (A3), the ratio $\eta_g$ weights larger values of $t$ relative to smaller ones. When $g(t)$ concentrates mass at small $t$-as occurs empirically when high-frequency words cluster near the origin-the moment ratio $\eta_g \to 0$. Consequently, the curvature correction in (4) vanishes, and the marginal $p(u)$ becomes approximately proportional to the intrinsic prior $h(u)$ alone.

This is the central observation: the geometric curvature bias (Corollary 2.2) that would otherwise enhance sampling along high-curvature directions is *attenuated* when the ambient radial density $g(t)$ concentrates at small $t$. High-curvature regions of the manifold, which the bias would naturally over-sample, become under-represented in the marginal distribution. The result is a loss of expressivity: the observed sample fails to reflect the intrinsic geometric structure of $\mathcal{M}$, and curvature-derived features are suppressed. Appendix C provides a visual illustration and an accessible introduction to these geometric concepts.

**Secondary effect: compression of intrinsic variation.** A secondary geometric effect reinforces the loss of expressivity. By (1), for fixed intrinsic radius $r$, high-curvature directions yield *smaller* ambient distances $t$. Consequently, intrinsic variation along curved directions is compressed into narrower ambient shells. Let $\Delta r = r_2 - r_1$ be an intrinsic radial increment; the corresponding ambient increment satisfies

$$\frac{\Delta t}{\Delta r} = 1 - \tfrac{1}{8}\kappa(u)r^2 + \mathcal{O}(r^3). \tag{5}$$

For directions with $\kappa(u) > 0$, the ratio $\Delta t/\Delta r < 1$, meaning equal intrinsic increments produce smaller ambient increments. When $g(t)$ concentrates at small $t$, this compression compounds the attenuation of curvature bias: not only is the bias itself suppressed, but the geometric locus where high-curvature points reside (smaller $t$ for given $r$) coincides

precisely with the region where sampling is densest, further obscuring the distinction between curved and flat directions.

**Interpretation.** Proposition 2.1 and Corollary 2.2 establish that curvature induces a directional bias at any fixed ambient radius through two complementary mechanisms: intrinsic curvature $\mathcal{R}(u)$ (Ricci curvature) governs how the Riemannian volume element distorts on $\mathcal{M}$, while extrinsic curvature $\kappa(u)$ (second fundamental form) controls how geodesics in $\mathcal{M}$ compress relative to ambient chords. Both effects make high-curvature directions geometrically over-represented. This bias is intrinsic to the manifold geometry (A1)-(A2) and, in the absence of radial sampling concentration, would enhance the visibility of curved regions in the embedding. However, assumption (A3) introduces a radial weighting $g(t)$ that, when concentrated at small $t$, drives the moment ratio $\eta_g \to 0$ and thereby attenuates *both* curvature contributions in the marginal distribution $p(u)$. The net effect is a loss of geometric expressivity: the observed sample under-represents high-curvature directions relative to what the intrinsic geometry would predict.

**Limitations.** The formal results derived above are inherently local: the chord-arc expansion (1) and its inverse (12) require $t$ to remain small relative to the reach $\tau$. Consequently, the quantitative predictions-particularly the $\mathcal{O}(t^4)$ corrections in squared distances and the $\mathcal{O}(t^2)$ corrections in densities-may not hold uniformly across the entire manifold. However, this locality is precisely aligned with the regime of interest: when $g(t)$ concentrates at small $t$, the mass of the observed sample lies within the validity range of the expansion. Thus, while the analysis may not fully characterize the global geometry of $\mathcal{M}$, it accurately describes the *observed* manifold as revealed by frequency-biased sampling. Under stronger regularity assumptions (e.g., larger reach, or condition on injectivity radius), the chord-arc comparison extends to larger neighborhoods, but the qualitative conclusion persists: radial concentration of the sampling density attenuates the curvature bias and reduces the geometric expressivity of the embedding.

### 2.4. Gradient Bias Toward Tangent Directions

The geometric analysis of Section 2.3 reveals that radial concentration attenuates curvature visibility in the *sampling* distribution. We now study a complementary *optimization* bias arising during training: gradient updates preferentially reinforce tangent directions over normal directions, creating a self-amplifying mechanism that further suppresses curvature-derived features.

**Tangent-normal decomposition.** Let $\mu \in \mathcal{M}$ be a reference point. For a sample $x \in \mathcal{M}$ close to $\mu$, write the centered representation as

$$x_c := x - \mu = v + n, \qquad v \in T_\mu\mathcal{M},\ n \in N_\mu\mathcal{M}.$$

The tangent component $v$ captures first-order displacement along $\mathcal{M}$, while the normal component $n$ encodes curvature-induced deviation from the tangent plane. Under (A1)-(A2), the local expansion of $\mathcal{M}$ yields [Lemma 3.3] (Berenfeld et al., 2021)

$$n = \tfrac{1}{2}\,\mathrm{II}_\mu(v,v) + \mathcal{O}(\|v\|^3), \qquad (6)$$

where $\mathrm{II}_\mu$ is the second fundamental form at $\mu$. Thus for $t := \|v\|$ small, we have $\|n\| = \mathcal{O}(t^2)$.

**Covariance structure.** Recall from (4) that when $\eta_g \to 0$, the marginal directional distribution $p(u)$ becomes approximately uniform on $S^{k-1} \subset T_\mu\mathcal{M}$. Writing $v = t\,u$ with $u$ drawn from this approximately uniform distribution, standard concentration arguments give

$$\mathbb{E}[vv^\top] = c_k\, t^2\, P_T, \qquad (7)$$

where $c_k > 0$ is a dimension-dependent constant. In contrast, the bounded curvature assumption (A2) with $\|\mathrm{II}\|_{\mathrm{op}} \leq C$ implies

$$\mathbb{E}[nn^\top] \preceq c_k\, C^2\, t^4\, P_N. \qquad (8)$$

Where $P_T$ and $P_N$ denote a $k$-**dimensional** orthogonal sub-projections onto the tangent space $T_\mu M$ and the normal space $N_\mu M$, respectively. The key observation is the *separation of scales*: tangent covariance is $\mathcal{O}(t^2)$, while normal covariance is $\mathcal{O}(t^4)$.

**Gradient structure for linear layers.** Consider a linear layer $W \in \mathbb{R}^{m \times d}$ acting on $x_c$. For a per-sample loss $\ell(Wx_c, y)$, define the backpropagated gradient $g(x) := \nabla_z \ell(z,y)|_{z=Wx_c}$ and let $G_{\mathrm{rms}} := \sqrt{\mathbb{E}[\|g\|^2]}$. The expected weight gradient decomposes as

$$\nabla_W \mathcal{L} = \mathbb{E}[g(x) \otimes x_c^\top] = \mathbb{E}[g \otimes v^\top] + \mathbb{E}[g \otimes n^\top].$$

Projecting onto tangent and normal subspaces and applying Cauchy-Schwarz together with (7)-(8):

$$\|\nabla_W \mathcal{L} \cdot P_T\|_F \leq G_{\mathrm{rms}}\sqrt{\mathbb{E}[\|v\|^2]} = \mathcal{O}(G_{\mathrm{rms}}\, t), \qquad (9)$$

$$\|\nabla_W \mathcal{L} \cdot P_N\|_F \leq G_{\mathrm{rms}}\sqrt{\mathbb{E}[\|n\|^2]} = \mathcal{O}(G_{\mathrm{rms}}\, t^2). \qquad (10)$$

The ratio of normal to tangent gradient magnitudes therefore scales as

$$\frac{\|\nabla_W \mathcal{L} \cdot P_N\|_F}{\|\nabla_W \mathcal{L} \cdot P_T\|_F} = \mathcal{O}(t). \qquad (11)$$

For small $t$, weight updates along normal directions are parametrically suppressed relative to tangent updates.

**Proposition 2.3** (Tangent gradient dominance). *Under (A1)-(A2), if the radial distribution concentrates at small $t$ (i.e., $\eta_g \ll 1$), then gradient updates to any linear layer satisfy* (11). *Consequently, weight components aligned with normal directions receive updates of order $\mathcal{O}(t)$ smaller than those aligned with tangent directions.*

**On random initialization.** Random initialization does not counteract this asymmetry. In high dimensions, randomly initialized weights are nearly orthogonal to any fixed low-dimensional subspace by concentration of measure. Moreover, standard initializations scale weights as $\mathcal{O}(d^{-1/2})$ (Glorot & Bengio, 2010), making random contributions weak compared to the data-driven tangent signal. Thus, noise neither preferentially excites nor stabilizes normal components during early training.

**Feedback amplification.** The tangent bias compounds through a feedback mechanism:

1. *Immediate bias:* Each gradient step $\Delta W = -\eta\, \nabla_W \mathcal{L}$ updates input-columns of $W$ predominantly within $T_\mu\mathcal{M}$.

2. *Reinforcement:* After such updates, forward activations depend more strongly on $v$ than on $n$; downstream predictive signals therefore correlate more with tangent features, causing future gradients to align even more with $T_\mu\mathcal{M}$.

This might creates a self-reinforcing loop: normal components are (i) geometrically small ($\|n\| \sim t^2$), (ii) generate weaker gradients ($\sim t^2$ vs. $\sim t$), and (iii) are progressively starved once tangent directions are amplified.

**Amplification through attention.** In transformer architectures, attention scores are bilinear: $s(x) = (W_Q x_c)^\top (W_K x_c) = x_c^\top M\, x_c$ for $M = W_Q^\top W_K$. Substituting $x_c = v + n$ with $\|v\| = t$ and $\|n\| = \mathcal{O}(t^2)$:

$$s(x) = v^\top M v + 2\, v^\top M n + n^\top M n,$$

yielding terms of order $\mathcal{O}(t^2)$, $\mathcal{O}(t^3)$, and $\mathcal{O}(t^4)$, respectively. Attention scores -and hence softmax weights- are dominated by tangent contributions. Since softmax exponentiates score differences, it further amplifies the gap, reducing attention sensitivity to normal-induced perturbations.

**Propagation through residual connections.** In a transformer block, the residual update takes the form $x^{(\ell+1)} = x^{(\ell)} + H^{(\ell)}(x^{(\ell)})$. During early training, $H^{(\ell)}$ is small and its parameters have begun aligning with tangent directions due to the gradient bias above. Consequently, $H^{(\ell)}(x^{(\ell)})$ is itself predominantly tangent-aligned. The residual connection preserves and incrementally reinforces the tangent component, while merely copying the normal component without amplification. Iterating across layers, this mechanism propagates tangent structure depth-wise, further diminishing the relative influence of normal features.

**Caveats.** This analysis is local and perturbative, valid for early training and small $t$. The mechanism may weaken when: (i) sampling extends to larger radii where $\|n\|$ is not

negligible; (ii) rare nonlinear interactions amplify small normal components; or (iii) learned normalization gains, multi-token attention patterns, or late-stage gradients restructure the representation space. Nevertheless, for high-frequency tokens-which by (A3) concentrate at small $t$, the tangent bias operates in its regime of validity.

**Implications for anisotropy.** The tangent gradient bias provides a mechanism linking the geometric observations of Section 2.3 to the empirically observed anisotropy in embedding spaces. High-frequency tokens, concentrated at small $t$, are most subject to tangent expansion; their dominant tangent subspace may then attract other tokens through shared layer parameters. While late-stage training can in principle restructure internal representations, the convergence of training loss indicates that late updates are small. The anisotropic geometry developed during early training may thus persist, propagated through the self-reinforcing dynamics described above.

# 3. Empirical Validation: Tangent-Aligned Gradient Anisotropy

The theory predicts that anisotropy should be preferentially organized by directions tangent to the locally sampled activation manifold and amplified by true-gradient dynamics. We test this claim directly by extracting, for each model, layer, and anchor token, a low-rank activation-derived subspace and then evaluating ordinary backpropagated true gradients against that same subspace across training.

**Activation-derived concept proxy and cross-model protocol.** Our geometric framework identifies the local tangent space $T_\mu \mathcal{M}$ as the relevant object, but this space is not directly observable. We therefore use a concept-based mechanistic-interpretability proxy (Fel et al., 2023a;b): for each model, layer, and anchor token, the pooled early activation cloud defines an activation-derived, low-rank concept subspace. Concretely, activations from early checkpoints are concatenated, centered, and used to fit an orthonormal basis $Q_T$ by low-rank PCA. Because the basis is extracted from activations rather than weights and is then tracked through training, it functions as a concept-like dictionary of dominant representation directions, without claiming equivalence to a specific concept-learning algorithm. $Q_T$ is fit once from pooled early activations and then kept fixed for evaluation in both the early and late phases.

We evaluate[1] EuroBERT-210m, EuroBERT-610m, moderncamembert-base, OLMo-1B, pythia-160m, pythia-410m, pythia-1b, Gaperon-1.5B, SmolLM2-360M, and SmolLM2-1.7B, spanning encoder-style

[1] https://github.com/Raphael-Bernas/
Revisiting-Anisotropy-In-Language-Transformers.
git

and decoder-style language models across scale. All models are tested on the same merged corpus built by subsampling from `allenai/c4` (English), `Salesforce/wikitext` (wikitext-103-raw-v1), `HuggingFaceFW/fineweb-edu` (sample-10BT), `ccdv/arxiv-summarization` (article field), and `wikimedia/wikipedia` (20231101.fr). Checkpoints are sampled uniformly from the first $30\%$ of training and the last $30\%$ (respectively *early* and *late*). The table reports the first block input/output together with the middle and last block inputs; for Pythia-like GPT-NeoX models, we provide also the pre-transformer first branch and the first true transformer-layer input.

**Gradient tests and displayed statistics.** For a selected token position $j$, let $x_j \in \mathbb{R}^D$ be the activation row, $\mu$ the fitted early-context mean, and $\delta_j \in \mathbb{R}^{D_{\text{out}}}$ the output-side backpropagated gradient row. With $x_j^c = x_j - \mu$, we form the centered true-gradient matrix

$$B = \sum_j \delta_j (x_j^c)^\top.$$

The first test asks whether gradient energy concentrates in the activation-derived tangent directions rather than in equally large normal directions. For a subspace basis $Q$, define

$$\mathcal{E}(Q) = \frac{\|BQ\|_F^2}{\dim(Q)}, \qquad R_{\text{energy}} = \frac{\mathcal{E}(Q_T)}{\mathcal{E}(Q_{N,\text{det}})},$$

where $Q_{N,\text{det}} \subset T^\perp$ is a deterministic matched-rank normal comparator. The accompanying null samples $S = 20$ random matched-rank normal subspaces and reports

$$p_{\text{energy}} = \frac{1 + \#\{s : \mathcal{E}(Q_N^{(s)}) \geq \mathcal{E}(Q_T)\}}{S + 1}.$$

Thus $R_{\text{energy}} > 1$ and small $p_{\text{energy}}$ indicate that true-gradient energy is unusually concentrated in the activation-derived tangent proxy.

The second test asks whether these same directions explain anisotropy rather than merely large variance. Let $\Sigma_B = B^\top B / D_{\text{out}}$, let $\widetilde{\Sigma}_B$ be its shrinkage-regularized version, and let $\text{Iso}^*(\widetilde{\Sigma}_B) \in [0,1]$ denote IsoScore* (Rudman & Eickhoff, 2024). We define

$$\Delta \text{Iso}_T^* = \text{Iso}^*(\widetilde{\Sigma}_{B \setminus Q_T}) - \text{Iso}^*(\widetilde{\Sigma}_B),$$

and analogously $\Delta \text{Iso}_{N,\text{det}}^*$ for the matched-normal comparator. The table reports

$$100 \frac{\Delta \text{Iso}_T^*}{\text{Iso}^*(\widetilde{\Sigma}_B)} \qquad / \qquad 100 \frac{\Delta \text{Iso}_{N,\text{det}}^*}{\text{Iso}^*(\widetilde{\Sigma}_B)},$$

together with the one-sided exact sign test across anchors,

$$p_{\text{Iso}, T>N} = \Pr[\text{Binom}(n, 1/2) \geq \#\{a : d_a > 0\}],$$

$$d_a = \overline{\Delta\mathrm{Iso}_T^{*(a)} - \Delta\mathrm{Iso}_{N,\mathrm{det}}^{*(a)}}.$$

Positive tangent contributions together with much smaller matched-normal values show that removing $Q_T$ recovers more isotropy than removing an equally large normal basis; values above 100 are possible when the baseline gradient covariance is strongly anisotropic and tangent removal leaves a nearly isotropic residual. The energy and IsoScore* statistics are complementary: the first establishes concentration of true-gradient energy in the activation-derived concept subspace, while the second shows that the same subspace specifically explains anisotropy. Further mechanistic-interpretability analyses of activation-space geometry and realized embedding updates are reported in Appendices D and E.

Table 1 summarizes the cross-model, cross-phase, and cross-layer results for this fixed early activation-derived subspace.

**Observations.** The pattern is strong and coherent across models. In the early phase, tangent directions typically capture substantially more gradient energy per dimension than matched normal directions, often by an order of magnitude or more, and the corresponding energy-null $p$-values are frequently at the Monte Carlo floor ($\frac{1}{S+1} = \frac{1}{21} \approx 4.8\mathrm{e}-2$). The effect is reduced in late training, but this later weakening does not make the early phase less important. Rather, it is precisely the phase in which the tangent-biased signal is strongest, and prior work shows that internal representations approach their final form early in training (Raghu et al., 2017), that many layers stabilise within the early portion of pretraining (Diehl Martinez et al., 2024b), and that gradient descent rapidly concentrates in a very small subspace (Gur-Ari et al., 2019). Taken together, these results are consistent with the view that anisotropy is imprinted primarily during the early regime, when tangent-biased gradients are strongest, and is then carried forward under smaller subsequent updates. The anisotropy test supports the same picture from a complementary angle: across models, phases, and layers, removing the tangent proxy usually improves isotropy more than removing a matched normal basis, and the one-sided sign test is often strongly significant across anchors. The recovered fraction is sometimes modest, and a few late-phase rows even become slightly negative, but this does not contradict the energy results: IsoScore* is sensitive to residual spectral imbalance and does not by itself identify a unique carrier subspace (see Appendix F), so tangent removal need not fully isotropize the residual when the remaining spectrum is still uneven or when the fixed early basis only approximates the later tangent directions. A depth trend is also visible, with earlier and middle layers usually showing the strongest effects, while final layers remain tangent-biased but are more exposed to objective-specific reshaping near the loss. Finally, the encoder models, especially EuroBERT, exhibit weaker alignment than the decoder models, in line with their weaker frequency-concentration signal; this is again consistent with the theory, which predicts that milder concentration should induce a weaker, not absent, tangent-bias mechanism.

## 4. Related Work

*Token-level syntactic geometry:* Structural probing shows that dependency-tree distances can be recovered from linear transforms of contextual embeddings (Hewitt & Manning, 2019), and geometric analyses of BERT report separable syntactic and semantic subspaces (Coenen et al., 2019). More recent studies move from global linear structure to local geometric characterization of token spaces, estimating intrinsic dimension and curvature and arguing for stratified, non-manifold structure (Robinson et al., 2026; 2025).

*Frequency effects and evolution of representation space:* Frequency leaves systematic geometric signatures in contextualized embeddings, distorting similarity geometry and affecting identifiability (Zhou et al., 2021). Degeneration phenomena are also tied to rare-token dynamics: rare-token gradients can drive global embedding drift and degrade representational quality (Biś et al., 2021; Yu et al., 2022). Related work further argues that frequency-aware geometric corrections can materially change how anisotropy should be interpreted and mitigated (Yokoi et al., 2024; Diehl Martinez et al., 2024a).

*Anisotropy in Transformer representations:* Anisotropy has been repeatedly documented in contextual embeddings across architectures (Ethayarajh, 2019). In generation settings, likelihood training with weight tying can collapse embeddings into a narrow cone (Gao et al., 2019). At the representation-analysis level, rogue or outlier dimensions can obscure representational quality (Timkey & van Schijndel, 2021), and there is evidence that anisotropy is not necessarily the sole cause of poor semantic behavior in BERT embeddings, motivating a more nuanced view that includes linguistic biases (Fuster Baggetto & Fresno, 2022). More recently, work on isotropy measurement has shown that several commonly used metrics are brittle (Rudman et al., 2022); other results argue that strict isotropy can conflict with clustered embedding spaces and linear classification objectives (Mickus et al., 2024). In the same vein, stable anisotropic regularization explicitly controls isotropy during training and reports that preserving or increasing anisotropy can improve downstream performance in many settings (Rudman & Eickhoff, 2024).

*Geometric and differential-geometric lenses:* Classical post-processing approaches that remove dominant directions ("all-but-the-top") highlight how a small set of principal components can dominate embedding geometry (Mu & Viswanath, 2018). Beyond global PCA-style corrections,

*Table 1.* Cross-model tangent-alignment summary for the shared early-context tangent fit, transposed and split into two stacked five-model panels. Early and late statistics appear as row blocks. $E$-r is the matched-normal energy ratio; $E$-null $p$, the Monte Carlo normal-space null; $\Delta I^*\%$, $100 \times \Delta \mathrm{Iso}^*/\mathrm{Iso}^*_{base}$, reported separately for tangent (T) and matched-normal (N) removal; and Iso $T > N\ p$, the one-sided sign test for $\Delta \mathrm{Iso}^*_T > \Delta \mathrm{Iso}^*_{N,\mathrm{det}}$. Starred layers mark the primary transformer-layer test; effect-size rows are bold and p-values plain. For Pythia, $E^0_{in}/E^0_{out}$ are the embedding input and post-embedding-dropout pre-transformer probes, and $T^1_{in}, T^1_{out}, T^m, T^\ell$ denote first-layer input, first-layer output, middle-layer input, and last-layer input.

| Metric | EuroBERT-210m | | | | EuroBERT-610m | | | | Gaperon-1.5B | | | | moderncamembert-base | | | | OLMo-1B | | | |
|---|---|---|---|---|---|---|---|---|---|---|---|---|---|---|---|---|---|---|---|---|
| | $T^1_{in}$* | $T^1_{out}$ | $T^m$ | $T^\ell$ | $T^1_{in}$* | $T^1_{out}$ | $T^m$ | $T^\ell$ | $T^1_{in}$* | $T^1_{out}$ | $T^m$ | $T^\ell$ | $T^1_{in}$* | $T^1_{out}$ | $T^m$ | $T^\ell$ | $T^1_{in}$* | $T^1_{out}$ | $T^m$ | $T^\ell$ |
| **EARLY** | | | | | | | | | | | | | | | | | | | | |
| $E_r$ | **7.42** | **53.1** | **20.6** | **18.6** | **10.7** | **68.6** | **21.3** | **20.6** | **239** | **1.4e3** | **46.8** | **43.3** | **225** | **64** | **66.2** | **30.1** | **364** | **1.2e3** | **58.7** | **66.1** |
| $P_{Enull}$ | 4.8e-2 | 4.8e-2 | 4.8e-2 | 4.8e-2 | 4.8e-2 | 4.8e-2 | 9.1e-2 | 8.7e-2 | 4.8e-2 | 4.8e-2 | 4.8e-2 | 4.8e-2 | 4.8e-2 | 4.8e-2 | 4.8e-2 | 4.8e-2 | 4.8e-2 | 4.8e-2 | 4.8e-2 | 4.8e-2 |
| $\Delta I^\star_T\%$ | **1.92** | **3.32** | **39.8** | **38.4** | **1.06** | **2.98** | **24.1** | **22.7** | **62.1** | **89.6** | **3.97** | **4.15** | **5.68** | **8.6** | **37.1** | **9.51** | **85.1** | **104** | **16.1** | **11.3** |
| $\Delta I^\star_N\%$ | **-0.157** | **-0.026** | **-0.398** | **-0.373** | **-0.217** | **-0.015** | **-0.207** | **-0.165** | **-0.189** | **-0.052** | **-0.043** | **-0.05** | **-0.006** | **-0.062** | **-0.233** | **-0.09** | **-0.186** | **-0.066** | **-0.098** | **-0.068** |
| $p_>^{\Delta I^\star}$ | 1.1e-2 | 6.0e-8 | 6.0e-8 | 6.0e-8 | 1.1e-2 | 6.0e-8 | 6.0e-8 | 6.0e-8 | 6.0e-8 | 6.0e-8 | 7.6e-2 | 0.271 | 6.0e-8 | 1.5e-6 | 6.0e-8 | 1.4e-4 | 1.5e-6 | 6.0e-8 | 1.5e-6 | 1.4e-4 |
| **LATE** | | | | | | | | | | | | | | | | | | | | |
| $E_r$ | **4.05** | **117** | **2** | **1.66** | **9.27** | **256** | **2.33** | **1.6** | **77.5** | **608** | **40.4** | **29.5** | **419** | **102** | **337** | **110** | **26.9** | **565** | **21.8** | **27.1** |
| $P_{Enull}$ | 4.8e-2 | 4.8e-2 | 6.5e-2 | 0.167 | 4.8e-2 | 4.8e-2 | 5.9e-2 | 8.9e-2 | 4.8e-2 | 4.8e-2 | 4.8e-2 | 4.8e-2 | 4.8e-2 | 4.8e-2 | 4.8e-2 | 4.8e-2 | 4.8e-2 | 4.8e-2 | 4.8e-2 | 4.8e-2 |
| $\Delta I^\star_T\%$ | **1.74** | **8.06** | **1.95** | **1.09** | **2** | **11.7** | **1.77** | **1.13** | **12.2** | **37.9** | **9.59** | **1.12** | **50.7** | **27.4** | **190** | **46.7** | **5.22** | **91.5** | **2.28** | **1.55** |
| $\Delta I^\star_N\%$ | **0.081** | **-0.03** | **-0.084** | **0.051** | **0.017** | **-0.022** | **-0.011** | **0.028** | **0.004** | **-0.029** | **0.044** | **-0.01** | **-0.034** | **-0.152** | **-0.373** | **-0.228** | **-0.054** | **-0.049** | **-0.041** | **-0.052** |
| $p_>^{\Delta I^\star}$ | 6.0e-8 | 6.0e-8 | 6.0e-8 | 1.8e-5 | 6.0e-8 | 6.0e-8 | 1.5e-6 | 1.8e-5 | 3.3e-3 | 1.8e-5 | 3.3e-3 | 0.419 | 6.0e-8 | 6.0e-8 | 6.0e-8 | 1.5e-6 | 3.3e-3 | 1.5e-6 | 3.2e-2 | 0.271 |

| Metric | pythia-1b | | | | | pythia-160m | | | | | pythia-410m | | | | | SmolLM2-1.7B | | | | SmolLM2-360M | | | |
|---|---|---|---|---|---|---|---|---|---|---|---|---|---|---|---|---|---|---|---|---|---|---|---|
| | $E^0_{in}$ | $E^0_{out}$ | $T^1_{in}$* | $T^m$ | $T^\ell$ | $E^0_{in}$ | $E^0_{out}$ | $T^1_{in}$* | $T^m$ | $T^\ell$ | $E^0_{in}$ | $E^0_{out}$ | $T^1_{in}$* | $T^m$ | $T^\ell$ | $T^1_{in}$* | $T^1_{out}$ | $T^m$ | $T^\ell$ | $T^1_{in}$* | $T^1_{out}$ | $T^m$ | $T^\ell$ |
| **EARLY** | | | | | | | | | | | | | | | | | | | | | | | |
| $E_r$ | **4.0e6** | **1.2e7** | **323** | **44.8** | **44.9** | **6.5e7** | **5.0e8** | **545** | **44.3** | **49** | **4.5e6** | **1.0e7** | **232** | **38.1** | **25.2** | **181** | **804** | **46** | **64.6** | **1.4e4** | **1.6e4** | **41.1** | **57.3** |
| $P_{Enull}$ | 4.8e-2 | 4.8e-2 | 4.8e-2 | 4.8e-2 | 4.8e-2 | 4.8e-2 | 4.8e-2 | 4.8e-2 | 4.8e-2 | 4.8e-2 | 4.8e-2 | 4.8e-2 | 4.8e-2 | 4.8e-2 | 4.8e-2 | 4.8e-2 | 4.8e-2 | 4.8e-2 | 4.8e-2 | 4.8e-2 | 4.8e-2 | 4.8e-2 | 4.8e-2 |
| $\Delta I^\star_T\%$ | **5.8e3** | **3.0e4** | **85.4** | **6.2** | **6.03** | **1.4e4** | **3.4e4** | **161** | **28.8** | **32.1** | **1.3e3** | **1.7e3** | **81** | **10.4** | **7.18** | **43.7** | **57.7** | **9.62** | **7.24** | **497** | **308** | **16.3** | **11.2** |
| $\Delta I^\star_N\%$ | **-1.0e-4** | **-8.4e-5** | **-0.198** | **-0.046** | **-0.055** | **-8.4e-5** | **-1.2e-4** | **-0.459** | **-0.4** | **-0.382** | **-3.8e-5** | **-5.9e-5** | **-0.318** | **-0.222** | **-0.186** | **-0.171** | **-0.039** | **-0.041** | **-0.07** | **-0.067** | **-0.025** | **-0.168** | **-0.126** |
| $p_>^{\Delta I^\star}$ | 6.0e-8 | 6.0e-8 | 6.0e-8 | 1.1e-2 | 3.3e-3 | 6.0e-8 | 6.0e-8 | 6.0e-8 | 6.0e-8 | 6.0e-8 | 6.0e-8 | 6.0e-8 | 1.5e-6 | 7.7e-4 | 7.6e-2 | 3.3e-3 | 7.7e-4 | 3.2e-2 | 1.1e-2 | 1.8e-5 | 1.4e-4 | 1.4e-4 | 7.6e-2 |
| **LATE** | | | | | | | | | | | | | | | | | | | | | | | |
| $E_r$ | **3.9e3** | **4.7e3** | **96.7** | **36.9** | **39.3** | **794** | **1.8e3** | **76.2** | **19.7** | **15.1** | **1.1e3** | **1.8e3** | **75.2** | **31.2** | **17.5** | **33.3** | **204** | **19.7** | **18.3** | **183** | **182** | **14.9** | **10.8** |
| $P_{Enull}$ | 4.8e-2 | 4.8e-2 | 4.8e-2 | 4.8e-2 | 4.8e-2 | 4.8e-2 | 4.8e-2 | 4.8e-2 | 4.8e-2 | 4.8e-2 | 4.8e-2 | 4.8e-2 | 4.8e-2 | 4.8e-2 | 4.8e-2 | 4.8e-2 | 4.8e-2 | 4.8e-2 | 4.8e-2 | 4.8e-2 | 4.8e-2 | 4.8e-2 | 4.8e-2 |
| $\Delta I^\star_T\%$ | **-0.01** | **-0.01** | **19.3** | **4.5** | **5.39** | **-0.003** | **-0.005** | **18** | **6.36** | **-2.49** | **-0.002** | **-0.004** | **7.63** | **6.4** | **-2.78** | **9.73** | **18.5** | **4.9** | **3.63** | **6.65** | **4.27** | **13.6** | **4.19** |
| $\Delta I^\star_N\%$ | **-7.9e-5** | **-9.9e-5** | **-0.043** | **-0.021** | **-0.055** | **-4.8e-5** | **-1.5e-5** | **-0.144** | **-0.187** | **-0.29** | **-5.5e-5** | **-9.8e-5** | **-0.135** | **-0.156** | **-0.155** | **-0.02** | **-0.004** | **0.007** | **-0.023** | **-0.013** | **-0.016** | **-0.195** | **-0.105** |
| $p_>^{\Delta I^\star}$ | 1 | 1 | 1.8e-5 | 3.3e-3 | 1.1e-2 | 1 | 1 | 3.3e-3 | 3.3e-3 | 0.924 | 1 | 1 | 1.1e-2 | 7.7e-4 | 0.924 | 1.4e-4 | 3.3e-3 | 1.4e-4 | 7.7e-4 | 1.5e-6 | 1.8e-5 | 6.0e-8 | 1.4e-4 |

recent work estimates local invariants such as dimension and Ricci curvature and argues that token spaces behave like stratified manifolds with nontrivial curvature (Robinson et al., 2026; 2025; Li & Sarwate, 2025). These results motivate studying anisotropy via local geometry rather than only global covariance statistics.

*Mechanistic interpretability during training (non post hoc):* Most mechanistic interpretability work is performed on trained checkpoints; fewer papers track circuit formation *throughout* training. Notable exceptions analyze the emergence and dependencies of induction-head circuits by intervening on activations across training (Singh et al., 2024), and identify circuit emergence in in-context learning (Minegishi et al., 2025). This setting remains comparatively underexplored, which motivates using mechanistic interpretability as a genuinely dynamic tool rather than only as a post hoc diagnostic.

# 5. Conclusion

This work has provided geometric arguments linking frequency bias to geometric bias, and in doing so has offered an intuition for how anisotropy may emerge from progressive concentration in lower-dimensional directions during training. This perspective further supports a view already present in recent literature, namely that anisotropy should not be reduced to a pathological feature of Transformer representations, but may also be understood as an implicit structural bias, and perhaps even as a form of natural regularization that effectively shifts an overparameterized problem toward a more underparameterized regime compatible with improved generalization. The mechanism described here is purely local and concerns the early phase of training, precisely when convergence is least stable; yet this is also the phase in which the broad organization of the representation space appears to settle, suggesting that it may already capture an important part of how anisotropy is formed. At the same time, such a local account does not fully explain how the gradient signal induced by the loss subsequently reshapes and sharpens these representations, reinforcing particular directions more than others, possibly because they support clustered and linearly useful organization. Clarifying this later directional selection, and more generally the interaction between early geometric concentration and loss-driven gradient structure, appears to be a natural continuation of the present work.

## Acknowledgements

All of the computational experiments were conducted using resources from the Ruche platform of the "Mésocentre" computing center of Université Paris-Saclay, Centrale-Supélec and École Normale Supérieure Paris-Saclay, supported by CNRS and Région Île-de-France. This research was supported by the LIAGORA Labcom (ANR-24-LCV2-0014), funded by the French National Research Agency (ANR).

This work has also benefited from the AI Cluster ANITI and the research programs DEEL[2] and FOR[3]. ANITI is funded by the France 2030 program under the Grant agreement n°ANR-23-IACL-0002. DEEL and FOR are integrative programs of the AI Cluster ANITI, designed and operated jointly with IRT Saint Exupéry, with the financial support from its industrial and academic partners and the France 2030 program under the Grant agreement n°ANR-10-AIRT-01.

We are especially grateful to **Gabriel Merlin** for carefully proofreading the mathematical statements and for pointing us to the exact series expansion formula of the Riemannian volume element. We also warmly thank **Alexis-Raja Brachet** for his many helpful comments and suggestions.

## Impact Statement

This work presents primarily theoretical analysis and discussion of learning dynamics in transformer architectures during training. The research focuses on understanding the geometric and topological properties of learned representations across training checkpoints. Importantly, this work does not propose specific training procedure modifications, does not advocate for particular architectural changes, and does not explicitly address potential weaknesses or failure modes of transformer models that could have direct practical implications. As such, the findings constitute foundational research into neural network learning dynamics and are unlikely to have immediate or direct societal impact. Any future applications building upon these theoretical insights would require separate assessment of their potential societal implications.

## Use of Large Language Models

Large Language Models, specifically Claude Sonnet 4.5, were used in the preparation of this manuscript primarily to enhance readability and improve clarity of presentation. The LLM was not involved in experimental design, did not produce the base experimental codebase, did not interpret experimental results, and did not derive the theoretical contributions presented in this work. All scientific content, including hypothesis formulation, experimental methodology, data analysis, and theoretical conclusions, were developed entirely by the authors. The LLM's role was strictly limited to code optimization and debugging assistance, as well as syntactic verification and language refinement of the manuscript text.

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

## A. Proof of Proposition 2.1

We invert (1) to express $r^2$ as a function of $t$. Writing $r^2 = t^2 + \delta$ for some correction $\delta$ and substituting into (1):

$$t^2 = (t^2 + \delta) - \tfrac{1}{12}\kappa(u)(t^2 + \delta)^2 + \mathcal{O}(t^5).$$

Expanding and matching powers of $t$, we find $\delta = \tfrac{1}{12}\kappa(u)t^4 + \mathcal{O}(t^5)$, yielding the inverse relation

$$r^2 = t^2 + \tfrac{1}{12}\,\kappa(u)\,t^4 + \mathcal{O}(t^5). \tag{12}$$

Subtracting two instances of (12) for directions $u$ and $u'$ at the same ambient distance $t$ establishes the result.

## B. Proof of Corollary 2.2.

Taking the square root of (12) and expanding:

$$r(t, u) = t + \tfrac{1}{24}\kappa(u)t^3 + \mathcal{O}(t^4). \tag{13}$$

Differentiating (13) with respect to $t$:

$$\frac{\partial r}{\partial t} = 1 + \tfrac{1}{8}\kappa(u)t^2 + \mathcal{O}(t^3).$$

For the Ricci term, using $r^2 = t^2 + \tfrac{1}{12}\kappa(u)t^4 + \mathcal{O}(t^5)$ from (12):

$$1 + \mathcal{R}(u)\tfrac{r^2}{6} = 1 + \mathcal{R}(u)\tfrac{t^2}{6} + \mathcal{O}(t^3).$$

Expanding $r^{k-1} = t^{k-1}(1 + \tfrac{1}{24}\kappa(u)t^2)^{k-1} = t^{k-1}(1 + \tfrac{k-1}{24}\kappa(u)t^2 + \mathcal{O}(t^3))$, and substituting all factors into (2):

$$p(t, u) \propto g(t)\,t^{k-1}\,h(u) \cdot \left(1 + \mathcal{R}(u)\tfrac{t^2}{6}\right)$$
$$\times \left(1 + \tfrac{k-1}{24}\kappa(u)t^2\right)\left(1 + \tfrac{1}{8}\kappa(u)t^2\right).$$

Expanding to second order (noting $\tfrac{k-1}{24} + \tfrac{3}{24} = \tfrac{k+2}{24}$):

$$p(t, u) \propto g(t)\,t^{k-1}\,h(u)\left(1 + \mathcal{R}(u)\tfrac{t^2}{6} + \tfrac{k+2}{24}\kappa(u)t^2 + \mathcal{O}(t^3)\right). \tag{14}$$

The marginal density on $t$ is $p(t) = \int p(t, u)\,d\sigma(u)$. The conditional density $p(u \mid t) = p(t, u)/p(t)$ has $g(t)$ and $t^{k-1}$ cancel, yielding:

$$p(u \mid t) \propto h(u)\left(1 + \mathcal{R}(u)\tfrac{t^2}{6} + \tfrac{k+2}{24}\kappa(u)t^2 + \mathcal{O}(t^3)\right). \tag{15}$$

## C. Geometric Intuition for Curvature Bias

This appendix provides geometric intuition for the concepts underlying Section 2.3.

**Manifolds: the basic idea.** A *manifold* is a space that locally resembles Euclidean space but may have global curvature. Familiar examples include: (i) a cylinder, which locally looks like a plane but wraps around globally; (ii) the surface of a torus (donut shape), which has regions of positive and negative curvature. For world data, the manifold hypothesis posits that high-dimensional non-noise generated data vectors concentrate near a lower-dimensional curved surface $\mathcal{M} \subset \mathbb{R}^d$.

**Reach and local regularity.** The *reach* $\tau$ of a manifold measures how sharply it can curve. Formally, it is the largest distance such that every point within distance $\tau$ of $\mathcal{M}$ has a unique nearest point on $\mathcal{M}$. A small reach indicates tight curvature or self-proximity; a large reach indicates gentle curvature. Our analysis requires only that the reach be positive in local neighborhoods.

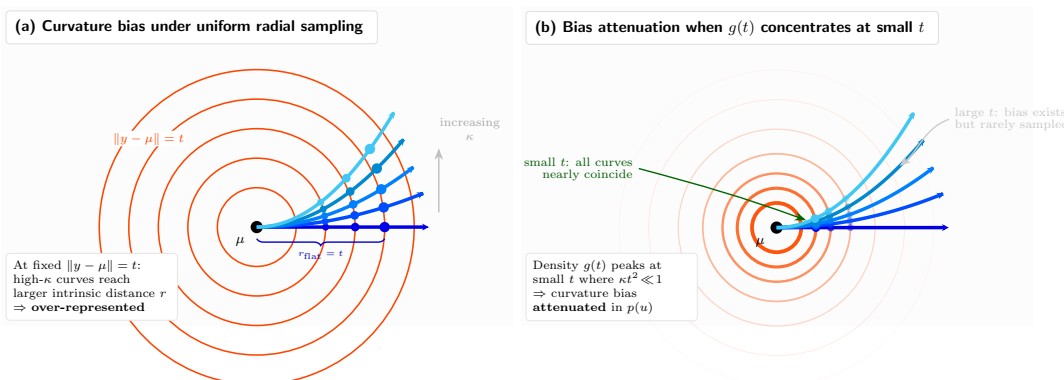

*Figure 2.* **Curvature bias and its attenuation.** (a) Under uniform radial sampling, directions with higher curvature $\kappa$ reach larger intrinsic distances $r$ at the same ambient radius $t$, causing over-representation of curved regions. (b) When the sampling density $g(t)$ concentrates at small $t$, all directions yield similar intrinsic displacements and the curvature bias is attenuated.

**Intrinsic vs. extrinsic distance.** Consider two points on the surface of a sphere. The *extrinsic* (ambient) distance is the straight-line chord through the interior; the *intrinsic* (geodesic) distance is the arc length along the sphere's surface. On flat surfaces these coincide, but on curved surfaces the arc length exceeds the chord length. This discrepancy is precisely what the chord-arc expansion (1) quantifies: $t^2 = r^2 - \frac{1}{12}\kappa(u)r^4 + \mathcal{O}(r^5)$, where $t$ is the chord (extrinsic) and $r$ is the arc (intrinsic).

**Consequences of non-global manifold structure.** Recent work shows that token embeddings may not form a single global smooth manifold but rather a stratified or piecewise structure (Robinson et al., 2026; 2025). This has practical consequences: global statistics (e.g., PCA across the entire vocabulary) may obscure local geometric phenomena. Our analysis therefore adopts a *local* manifold hypothesis, requiring smoothness only in semantic neighborhoods. This motivates the use of local estimators (intrinsic dimension, local curvature).

**Sampling on a manifold vs. ambient space.** A probability distribution on a manifold $\mathcal{M} \subset \mathbb{R}^d$ is fundamentally different from one on the full ambient space $\mathbb{R}^d$. If data truly lies on a lower-dimensional $\mathcal{M}$, then it has measure zero in $\mathbb{R}^d$, no "volume" in the ambient sense. Consequently, densities must be defined with respect to the intrinsic Riemannian measure on $\mathcal{M}$, not the Lebesgue measure on $\mathbb{R}^d$. When we sample according to an ambient radial density $g(t)$, we implicitly weight the intrinsic distribution by factors depending on how the manifold embeds into $\mathbb{R}^d$ - this is the origin of the curvature bias.

**Curvature bias: intuition.** Consider sampling points at a fixed ambient distance $t$ from a reference point $\mu$. On a flat manifold, all directions yield the same intrinsic distance. On a curved manifold, high-curvature directions "curve back" toward $\mu$, so reaching ambient distance $t$ requires traveling a longer intrinsic arc. At fixed $t$, high-curvature directions correspond to larger intrinsic displacements and are therefore over-represented in any sample conditioned on ambient radius. However, when sampling concentrates at small $t$ (as occurs for frequent tokens), the curvature correction becomes negligible and the bias is attenuated - this is the mechanism underlying our main theoretical result.

**Illustration.** Figure 2 visualizes these effects. Panel (a) shows that at uniform ambient radius, higher-curvature directions (lighter curves) reach larger intrinsic distances, leading to over-representation. Panel (b) shows that when the radial density concentrates at small radii, all curves nearly coincide and the curvature-induced bias is suppressed.

# D. Further Mechanistic-Interpretability Results During Training

The main paper uses true gradients as the primary empirical validation of tangent-aligned anisotropy. In addition, we carried out a broader mechanistic-interpretability study of activation-space geometry during training in order to observe more precisely how anisotropy is accompanied, inside the model, by a simplification of the active representation space. We report these results here as supplementary evidence: unlike the main text, they are not based on a direct tangent-versus-matched-normal gradient comparison, but they provide a complementary view of how the representation geometry itself evolves

toward a lower-dimensional and more stable organization.

The theoretical analysis of Sections 2.3-2.4 predicts that (i) radial concentration attenuates curvature visibility, and (ii) gradient updates preferentially reinforce tangent directions, producing a low-dimensional, approximately linear subspace that dominates the representation geometry. In this appendix, we study these predictions from the perspective of concept geometry across training.

**Concept directions as tangent-space proxies.** Our geometric framework posits that the tangent space $T_\mu \mathcal{M}$ captures the principal directions of variation under frequency-biased sampling. In practice, we do not observe the tangent space directly; instead, we employ concept-based analysis, a branch of mechanistic interpretability focused on decomposing neural representations and using them for human-interpretable analysis (Fel et al., 2023a;b), through extraction of a low-rank *concept directions* dictionary from intermediate activations. Critically, these concept directions are computed from pooled token activations, so high-frequency tokens contribute disproportionately to the learned dictionary. Under the tangent gradient dominance of Proposition 2.3, precisely these tokens drive weight updates along $T_\mu \mathcal{M}$, with $\mu$ an idealized form of the frequent-token embedding. Consequently, the extracted concept space inherits the structure of the dominant tangent directions: it represents the subspace that training has preferentially amplified. This motivates our use of concept geometry as a supplementary empirical surrogate for the theoretical tangent space. A complementary toy experiment verifying gradient alignment with the concept space is reported in Appendix G.

**Models and protocol.** We study three Transformer architectures of comparable scale: EuroBERT-210M (encoder), Pythia-410M (decoder), and SmolLM2-360M (decoder). This selection enables comparison across encoder and decoder paradigms while keeping dense checkpoint coverage throughout training. Each model provides multiple intermediate checkpoints, grouped into step ranges in Table 2 to track representation dynamics. For each checkpoint and tracked Transformer block, we extract MLP activations from a fixed subset of `Salesforce/wikitext` and build a pooled activation matrix $X \in \mathbb{R}^{M \times D}$. The aim is not to reproduce the main-paper gradient test, but rather to inspect how the internal activation geometry co-develops with anisotropy as training proceeds.

**Concept extraction.** We apply Semi-NMF (Ding et al., 2010; Parekh et al., 2024) to obtain concept directions: $X \approx ZH$ with $Z \geq 0$, where $H \in \mathbb{R}^{K \times D}$ contains concept vectors and $Z \in \mathbb{R}^{M \times K}$ the corresponding coefficients. The nonnegativity constraint on $Z$ naturally models the conic concentration typical of anisotropic representations while preserving signed concepts in $H$. All geometric statistics below are computed in this concept space. In that sense, the appendix probes whether the effective representational support becomes simpler and more aligned with a small set of dominant activation-derived directions as training progresses.

**Metrics.** We report the following quantities (cf. Table 2): *Linear anisotropy:* $\text{PCA}_{70\%}$, the number of components required to explain 70% of the variance; $\text{Sim}_{\text{PCA}}$, cosine similarity of leading eigenvectors across successive checkpoints. *Intrinsic dimension:* $\text{ID}_{2\text{nn}}$, the TwoNN estimate (Facco et al., 2017); $\text{ID}_{\text{Graph}}$, a graph-based estimator sensitive to curvature and manifold structure (Costa & Hero, 2006). *Effective rank:* $r_k^{\text{eff}}$, the spectral-entropy rank of the concept space (Roy & Vetterli, 2007). *Intrinsic entropy:* $S_\mathcal{M}$, graph-based entropy on $\mathcal{M}$ (Costa & Hero, 2006; Sadrtdinov et al., 2025), used to track distributional evolution on the learned support. *Concept-weight alignment:* $c_W$, maximum correlation between the activation-derived concept basis at layer $\ell$ and a concept basis extracted from $W^{(\ell+1)}$, quantifying the extent to which weights organize around learned concept directions.

**Predictions and interpretation.** Our analysis predicts that training progressively favors tangent directions, inducing an anisotropic flattening: manifold-support estimates should remain low, or contract, while linear spectral proxies can remain high as variance spreads over many weak directions. We therefore expect a stable scale separation, typically $r_k^{\text{eff}} \gg \text{ID}_{2\text{nn}} \gtrsim \text{ID}_{\text{Graph}}$, where $\text{ID}_{\text{Graph}}$ reflects a global manifold-like support and $\text{ID}_{2\text{nn}}$ reflects locally active degrees of freedom at nearest-neighbor scale. In the present appendix, this separation is the key descriptive signature: if anisotropy is indeed induced by training concentrating updates along a restricted tangent-aligned set of directions, then the model should behave as though its actively used geometry becomes simpler than the ambient linear dimension would suggest. Consistent tangent-space preference should further yield increasing $\text{Sim}_{\text{PCA}}$, indicating stabilization of principal directions, and nontrivial $c_W$, indicating that learned weights organize around those dominant directions. Finally, $S_\mathcal{M}$ tracks the convergence of the concept distribution on $\mathcal{M}$: when $S_\mathcal{M}$ evolves, geometry should co-evolve; when it stabilizes, geometry-related metrics should plateau.

**Observations.** Across models, the entropy heatmap closely tracks the evolution of dimension estimators, suggesting entropy as a reliable proxy for monitoring the mutual-information structure of representations during training. Throughout

*Table 2.* **Layerwise geometry across training checkpoints (Semi-NMF concept decomposition).** Each cell reports the *first*, *intermediate*, and *last* transformer layers (top→bottom; indices omitted because model depths differ). Salmon shading is min-max normalized *per model and metric* (stronger = larger value). Last-layer entries for $c_W$ in the provided logs are shown as "-" since their is no weight to correlate with. In $PCA_{70\%}$, a value above the decomposition threshold is noted 101 (which implies a need for more than 100 features to explain 70% variance, thus a better estimation in those cases is the effective rank $r_k^{\text{eff}}$).

| | EuroBERT-210M | | | | | | | | Pythia-410M | | | | | | | | SmolLM2-360M | | | |
|---|---|---|---|---|---|---|---|---|---|---|---|---|---|---|---|---|---|---|---|---|
| Metric | 10k-60k | 70k-120k | 130k-180k | 190k-240k | 250k-300k | 310k-360k | 370k-420k | 430k-480k | 10k-22k | 26k-38k | 42k-54k | 58k-70k | 78k-90k | 94k-106k | 110k-122k | 126k-138k | 160k-640k | 800k-1.28M | 1.44M-1.92M | 2.08M-2.56M |
| | 22 | 22 | 22 | 24 | 23 | 25 | 24 | 24 | 19 | 24 | 22 | 21 | 19 | 21 | 20 | 19 | 18 | 18 | 17 | 15 |
| | 21 | 10 | 7 | 9 | 10 | 10 | 9 | 12 | 37 | 27 | 28 | 25 | 22 | 23 | 25 | 24 | 19 | 25 | 27 | 23 |
| $ID_{\text{Graph}}$ | 22 | 24 | 24 | 23 | 20 | 20 | 21 | 22 | 6 | 5 | 4 | 4 | 4 | 4 | 5 | 5 | 11 | 7 | 5 | 5 |
| | 54.56 | 63.18 | 70.28 | 65.69 | 68.02 | 63.79 | 62.41 | 67.51 | 52.24 | 49.32 | 49.04 | 44.80 | 53.78 | 46.90 | 49.64 | 44.94 | 46.82 | 45.02 | 45.40 | 43.69 |
| | 52.08 | 33.19 | 23.93 | 28.18 | 32.23 | 31.47 | 32.26 | 34.10 | 76.46 | 74.92 | 68.65 | 65.89 | 64.83 | 62.67 | 64.69 | 68.03 | 73.38 | 72.84 | 76.89 | 63.82 |
| $ID_{\text{2nn}}$ | 60.58 | 57.36 | 52.52 | 49.62 | 47.95 | 46.67 | 42.93 | 45.47 | 16.07 | 15.48 | 15.13 | 11.93 | 11.36 | 12.21 | 11.81 | 11.56 | 30.43 | 21.72 | 19.09 | 15.30 |
| | 76 | 90 | 98 | 96 | 95 | 96 | 94 | 93 | 88 | 91 | 93 | 94 | 95 | 95 | 96 | 96 | 49 | 49 | 48 | 53 |
| | 25 | 1 | 1 | 1 | 1 | 17 | 1 | 1 | 101 | 101 | 101 | 101 | 101 | 101 | 101 | 101 | 101 | 101 | 101 | 101 |
| $PCA_{70\%}$ | 56 | 69 | 30 | 43 | 32 | 25 | 30 | 34 | 1 | 1 | 1 | 1 | 1 | 1 | 1 | 1 | 1 | 1 | 1 | 1 |
| | 166.7 | 194.5 | 202.2 | 198.0 | 198.6 | 196.4 | 195.0 | 196.9 | 141.5 | 136.5 | 133.6 | 134.0 | 131.7 | 129.7 | 129.3 | 128.0 | 119.2 | 105.7 | 104.9 | 87.6 |
| | 164.0 | 18.9 | 3.0 | 3.1 | 4.3 | 4.1 | 8.8 | 7.2 | 261.4 | 243.9 | 234.4 | 223.6 | 221.3 | 209.7 | 207.1 | 204.4 | 235.4 | 243.3 | 238.3 | 213.8 |
| $r_k^{\text{eff}}$ | 210.7 | 187.5 | 165.4 | 134.4 | 124.4 | 118.8 | 99.8 | 119.3 | 4.4 | 2.2 | 1.8 | 1.8 | 1.8 | 1.9 | 2.3 | 2.2 | 30.5 | 2.3 | 1.6 | 1.9 |
| | 0.514 | 0.851 | 0.810 | 0.852 | 0.897 | 0.847 | 0.571 | 0.825 | 0.956 | 0.985 | 0.988 | 0.991 | 0.995 | 0.997 | 0.998 | 0.998 | 0.935 | 0.971 | 0.964 | 0.922 |
| | 0.348 | 0.596 | 0.663 | 0.578 | 0.568 | 0.419 | 0.244 | 0.567 | 0.887 | 0.903 | 0.952 | 0.951 | 0.932 | 0.960 | 0.983 | 0.984 | 0.612 | 0.740 | 0.874 | 0.884 |
| $Sim_{\text{PCA}}$ | 0.230 | 0.387 | 0.516 | 0.314 | 0.350 | 0.237 | 0.335 | 0.302 | 0.988 | 0.923 | 0.952 | 0.983 | 0.973 | 0.996 | 0.991 | 0.998 | 0.540 | 0.780 | 0.688 | 0.627 |
| | 0.171 | 0.185 | 0.171 | 0.177 | 0.165 | 0.172 | 0.178 | 0.166 | 0.157 | 0.164 | 0.154 | 0.153 | 0.147 | 0.145 | 0.157 | 0.153 | 0.131 | 0.125 | 0.119 | 0.135 |
| | 0.176 | 0.171 | 0.159 | 0.163 | 0.146 | 0.161 | 0.173 | 0.161 | 0.154 | 0.176 | 0.158 | 0.150 | 0.159 | 0.175 | 0.160 | 0.155 | 0.184 | 0.181 | 0.151 | 0.146 |
| $c_W$ | - | - | - | - | - | - | - | - | - | - | - | - | - | - | - | - | - | - | - | - |
| | 55.5 | 40.6 | 37.0 | 38.0 | 36.6 | 38.2 | 37.5 | 37.2 | 74.4 | 90.8 | 84.7 | 80.3 | 73.1 | 77.5 | 73.3 | 69.9 | 91.4 | 92.2 | 88.9 | 78.4 |
| | 58.4 | 27.0 | 16.8 | 18.7 | 19.9 | 19.8 | 19.4 | 22.3 | 120.7 | 95.8 | 97.8 | 88.8 | 78.5 | 79.4 | 82.5 | 78.9 | 96.3 | 118.9 | 126.1 | 106.8 |
| $S_{\mathcal{M}}$ | 65.6 | 52.0 | 45.8 | 40.2 | 35.0 | 34.4 | 34.3 | 35.6 | 29.8 | 25.6 | 20.8 | 21.0 | 20.3 | 20.4 | 24.7 | 24.4 | 70.2 | 46.7 | 34.5 | 35.5 |

training, $ID_{\text{Graph}}$ is consistently smaller than $ID_{\text{2nn}}$, and both are markedly below $r_k^{\text{eff}}$, indicating representations supported on a comparatively low-dimensional structure while variance remains distributed across many linear directions. This is compatible with tangent-space amplification together with reduced local "thickness", that is, fewer locally active degrees of freedom beyond the manifold-like support. Layerwise trends further align with architecture: the encoder shows pronounced intermediate-layer compression, whereas decoders exhibit an expand-compress pattern with strong final-layer contraction for prediction. In parallel, $Sim_{\text{PCA}}$ is near-saturated in decoders, and comparatively higher in early encoder layers, consistent with stabilized principal directions once the geometry has converged, while $c_W$ indicates persistent, non-negligible alignment between MLP weights and dominant concept directions. Overall, the coupled evolution of $ID_{\text{Graph}}$, $ID_{\text{2nn}}$, $r_k^{\text{eff}}$, and $S_{\mathcal{M}}$ is consistent with the anisotropic-flattening mechanism and provides a more internal mechanistic picture of how anisotropy is accompanied by a simplification of the active representational space.

# E. Embedding-Update Enrichment by Token Frequency

The true-gradient results in the main paper provide the cleanest test of the tangent-alignment mechanism because they evaluate the loss-derived backpropagated signal directly. As a complementary analysis, we also examine the realized parameter evolution of the embedding matrix itself by measuring checkpoint-to-checkpoint updates $dW = W_{t+n} - W_t$, where $n$ is fixed by the available checkpoint granularity. This appendix asks whether the same tangent preference remains visible in the effective update, not only in the reconstructed gradient.

**Update-level tangent and normal enrichments.** For a tracked token $i$, let $e_i(1), \ldots, e_i(T) \in \mathbb{R}^D$ denote its embedding row across checkpoints in a given phase, let $\mu_i$ be the trajectory centroid, and let the centered trajectory matrix be formed from $e_i(t) - \mu_i$. We extract a low-rank tangent basis $Q_T^{(i)}$ by PCA/SVD on this trajectory cloud and use its orthogonal complement as the normal space. For each update row $\Delta e_i = e_i(t + n) - e_i(t)$, we measure the dimension-normalized energy captured by the tangent and normal components and then normalize each quantity by its random-subspace baseline, so that value 1 corresponds to chance level. The reported statistics are therefore tangent and normal enrichments relative to random, computed on the effective update $B = dW$.

**Reporting protocol.** Figure 3 shows these enrichments as a function of token frequency across models. Tokens are grouped into rarity bins, plotted against $\log_{10}(\text{freq})$, and color-coded by frequency group in order to make the frequency dependence explicit. Table 3 complements this aggregate view with exact values for representative tracked tokens, together with their empirical frequencies, in the early and stable regimes.

*Table 3.* **Token-level embedding-update enrichments.** For each model we report representative tracked tokens spanning the empirical frequency range, together with their frequencies and the corresponding tangent and normal enrichment ratios computed on the embedding update $dW$. Each ratio is normalized by its random-subspace baseline, so that 1 corresponds to chance level. Token selection is guided by the early regime, and the same tokens are then reused in the stable columns. (*) SmolLM2-360m only has 16 available checkpoints starting in the late training phase.

| Model | Token | Freq. | Early | | Stable | |
|---|---|---|---|---|---|---|
| | | | **Tangent** | **Normal** | **Tangent** | **Normal** |
| **EuroBERT-210m** | of | 2.40e-02 | 33.27 | 0.99 | 3.72 | 1.00 |
| | her | 1.63e-03 | 7.40 | 1.00 | 3.53 | 1.00 |
| | lines | 1.16e-04 | 2.79 | 1.00 | 2.24 | 1.00 |
| | jam | 6.45e-06 | 3.26 | 1.00 | 2.76 | 1.00 |
| | chat | 2.15e-06 | 3.63 | 1.00 | 2.02 | 1.00 |
| **EuroBERT-610m** | the | 4.58e-02 | 28.24 | 1.00 | 1.94 | 1.00 |
| | after | 1.11e-03 | 6.66 | 1.00 | 1.48 | 1.00 |
| | Christmas | 1.38e-04 | 4.02 | 1.00 | 1.36 | 1.00 |
| | streaming | 2.15e-06 | 6.04 | 1.00 | 1.57 | 1.00 |
| | chat | 2.15e-06 | 7.16 | 1.00 | 1.37 | 1.00 |
| **ModernCamenBERT** | ##s | 1.48e-02 | 53.90 | 0.99 | 22.16 | 0.98 |
| | his | 2.46e-03 | 4.07 | 1.00 | 1.90 | 1.00 |
| | corn | 1.02e-04 | 3.10 | 1.00 | 2.05 | 1.00 |
| | lib | 2.41e-05 | 4.41 | 1.00 | 1.75 | 1.00 |
| | ##acha | 2.01e-06 | 3.98 | 1.00 | 2.18 | 1.00 |
| **OLMo-1b** | for | 5.62e-03 | 16.23 | 1.00 | 5.45 | 0.96 |
| | but | 1.31e-03 | 4.88 | 1.00 | 1.94 | 0.99 |
| | love | 1.49e-04 | 7.17 | 1.00 | 1.90 | 0.99 |
| | helping | 1.95e-05 | 13.61 | 1.00 | 1.96 | 0.99 |
| | robust | 4.33e-06 | 14.93 | 1.00 | 2.64 | 0.99 |
| **pythia-1b** | to | 1.58e-02 | 128.97 | 0.99 | 15.80 | 0.99 |
| | into | 9.57e-04 | 3.66 | 1.00 | 6.88 | 1.00 |
| | Canadian | 1.78e-04 | 3.01 | 1.00 | 5.29 | 1.00 |
| | gains | 1.08e-05 | 2.58 | 1.00 | 6.51 | 1.00 |
| | generator | 2.16e-06 | 3.27 | 1.00 | 5.45 | 1.00 |
| **pythia-160m** | the | 4.65e-02 | 58.52 | 0.98 | 1.99 | 0.98 |
| | also | 1.46e-03 | 4.08 | 1.00 | 2.04 | 0.99 |
| | poem | 1.21e-04 | 2.05 | 1.00 | 4.47 | 0.98 |
| | gains | 1.08e-05 | 1.77 | 1.00 | 3.26 | 0.98 |
| | Dual | 2.16e-06 | 1.89 | 1.00 | 3.66 | 0.99 |
| **pythia-410m** | to | 1.58e-02 | 27.85 | 1.00 | 14.59 | 0.99 |
| | It | 1.05e-03 | 2.92 | 1.00 | 4.89 | 1.00 |
| | signed | 1.43e-04 | 2.32 | 1.00 | 4.75 | 1.00 |
| | instinct | 8.66e-06 | 2.06 | 1.00 | 5.54 | 1.00 |
| | doctrine | 4.33e-06 | 2.55 | 1.00 | 4.44 | 1.00 |
| **SmolLM2-1.7b** | c | 2.22e-02 | 112.84 | 0.97 | 23.42 | 0.96 |
| | M | 2.49e-03 | 41.13 | 0.99 | 10.65 | 0.98 |
| | L | 1.64e-03 | 37.79 | 0.99 | 10.20 | 0.97 |
| | x | 1.33e-03 | 51.25 | 0.98 | 7.79 | 0.96 |
| | q | 5.32e-04 | 82.81 | 0.98 | 9.82 | 0.95 |
| **SmolLM2-360m** (*) | i | 5.29e-02 | - | - | 7.29 | 1.00 |
| | y | 1.10e-02 | - | - | 4.12 | 0.99 |
| | x | 1.33e-03 | - | - | 4.13 | 0.99 |
| | q | 5.32e-04 | - | - | 5.28 | 0.99 |
| | X | 6.73e-05 | - | - | 6.26 | 0.99 |

**Interpretation.** The update-level picture is highly consistent with the main-paper gradient analysis. Across models, tangent enrichment is systematically larger than normal enrichment, with the largest contrasts typically observed for the most frequent tokens. For several decoder-style models, early tangent enrichment reaches one or even two orders of magnitude above the random baseline, whereas normal enrichment stays close to 1 and is often slightly depleted. This is the same qualitative signature as in the true-gradient study: updates are organized preferentially along tangent directions rather than equally sized normal directions.

The temporal pattern is also informative. The contrast is generally stronger in the early regime and weaker in the stable one, which agrees with the theoretical picture that much of the representational organization is established relatively early and that later training proceeds with a smaller residual signal. Importantly, the "early" segment available here already lies

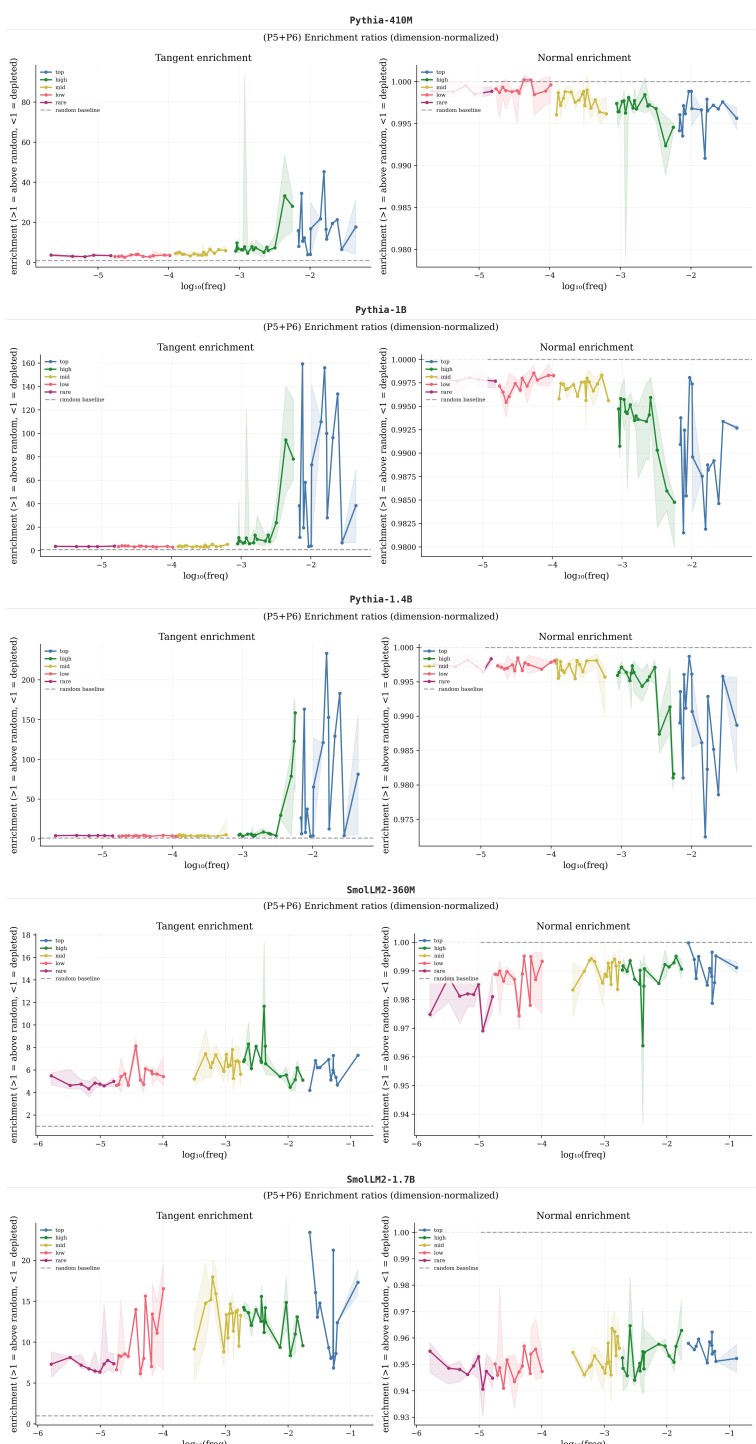

*Figure 3.* **Embedding-update tangent and normal enrichments across token frequency.** For each model, we compute tangent and normal enrichment ratios on the embedding update $dW = W_{t+n} - W_t$, normalized so that the random baseline is 1. The left panel in each model reports tangent enrichment; the right panel reports normal enrichment. Curves are plotted against $\log_{10}(\text{freq})$ and colored by token rarity group, which makes the frequency dependence visible while preserving within-group variability. Frequent tokens tend to exhibit the strongest tangent enrichment, whereas normal enrichment remains near, or slightly below, the random baseline. The effect is visibly weaker for the EuroBERT models, consistent with the milder frequency-concentration pattern in Figure 1.

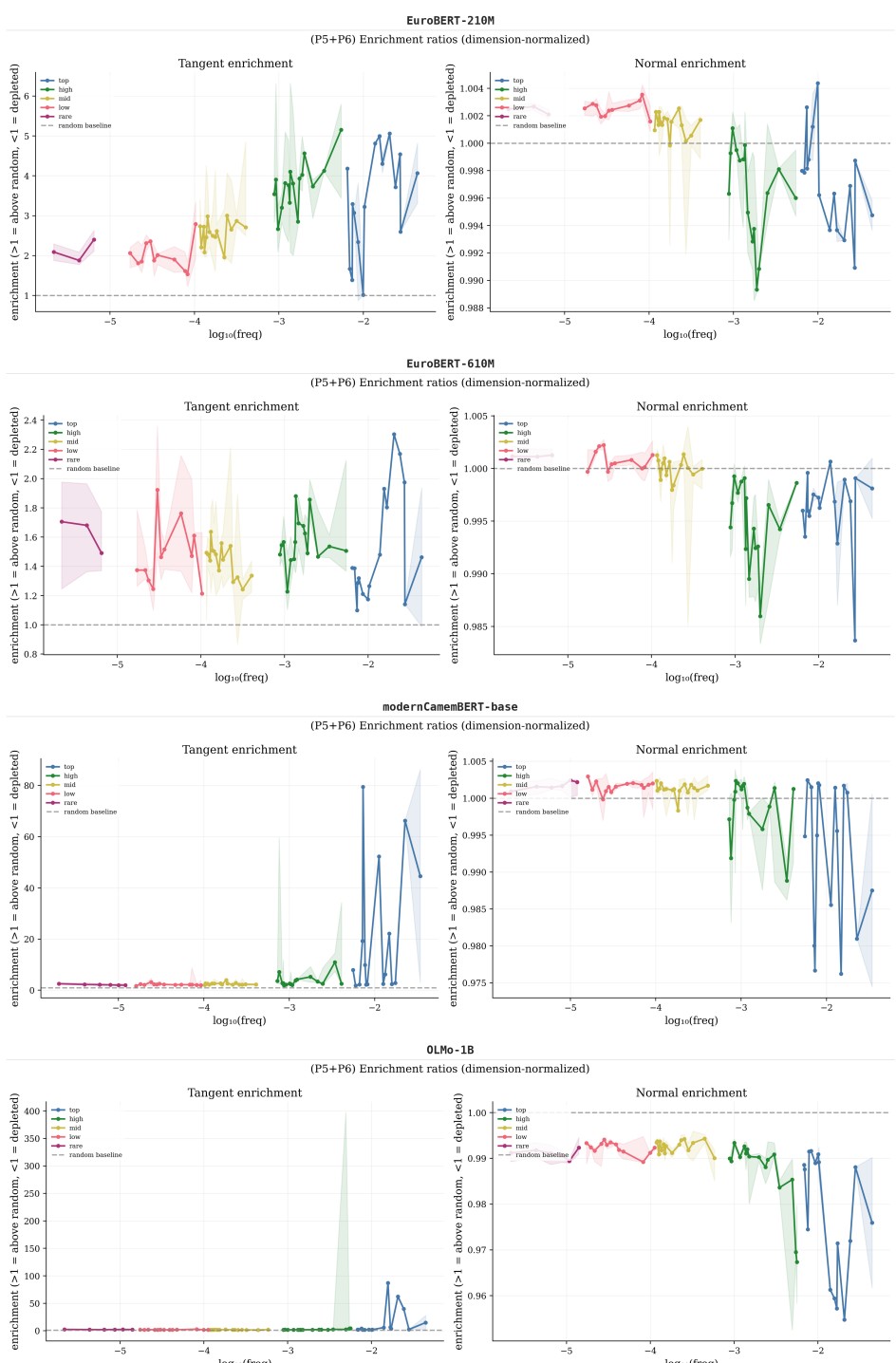

*Figure 3.* **Embedding-update tangent and normal enrichments across token frequency.** For each model, we compute tangent and normal enrichment ratios on the embedding update $dW = W_{t+n} - W_t$, normalized so that the random baseline is 1. The left panel in each model reports tangent enrichment; the right panel reports normal enrichment. Curves are plotted against $\log_{10}(\text{freq})$ and colored by token rarity group, which makes the frequency dependence visible while preserving within-group variability. Frequent tokens tend to exhibit the strongest tangent enrichment, whereas normal enrichment remains near, or slightly below, the random baseline. The effect is visibly weaker for the EuroBERT models, consistent with the milder frequency-concentration pattern in Figure 1.

well within the training trajectory rather than near initialization, so the observed tangent bias does not rely on a transient start-of-training effect. The weaker curves obtained for EuroBERT are also coherent with the earlier frequency-concentration analysis: these models show a milder contraction of frequent-token trajectories and, correspondingly, a smaller tangent-bias scale. This does not weaken the theoretical picture. Rather, it supports it: when the frequency-driven concentration effect is attenuated, the tangent-preference signal is attenuated as well. Even in this weaker regime, EuroBERT still shows tangent enrichment clearly above the matched normal baseline for frequent tokens, so it is best interpreted as a weaker instance of the same mechanism rather than a counterexample. Overall, the effective embedding updates provide a complementary validation that the parameter motion itself remains more aligned with tangent than with normal directions throughout the available training trajectory.

## F. Why IsoScore* Alone Does Not Identify the Carrier Subspace

IsoScore* is a useful summary statistic for anisotropy. It inherits from IsoScore the same normalize-distance-rescale construction, but applies it to the eigenspectrum of the shrunk covariance matrix rather than to the diagonal of the PCA-reoriented covariance (Rudman et al., 2022; Rudman & Eickhoff, 2024). This makes it compact, scale-free, and almost everywhere differentiable, which is precisely why it is attractive in optimization-oriented settings. Its limitation is different: it quantifies *how much* spectral anisotropy is present, but not *where* that anisotropy lives geometrically.

**Closed form.** Let $\lambda = (\lambda_1, \ldots, \lambda_d)$ be the nonnegative eigenvalues of the shrunk covariance matrix $\Sigma_\zeta$. Following Rudman & Eickhoff (2024), IsoScore* normalizes the spectrum as $\hat{\lambda} = \sqrt{d}\,\lambda/\|\lambda\|_2$, defines the isotropy defect by the Euclidean distance from $\mathbf{1}$, and then applies the same occupancy and affine rescaling used in IsoScore (Rudman et al., 2022). A short algebraic simplification gives a closed form depending only on the $\ell_1$ and $\ell_2$ norms of the spectrum. Indeed, because $\|\hat{\lambda}\|_2^2 = d$,

$$\|\hat{\lambda} - \mathbf{1}\|_2^2 = \|\hat{\lambda}\|_2^2 + \|\mathbf{1}\|_2^2 - 2\langle\hat{\lambda}, \mathbf{1}\rangle = 2d - 2\sum_{i=1}^{d} \hat{\lambda}_i,$$

so the defect term depends only on $\sum_i \hat{\lambda}_i$. Using $\sum_i \hat{\lambda}_i = \sqrt{d}\,\|\lambda\|_1/\|\lambda\|_2$, the occupancy step collapses, after substitution, to

$$\text{IsoScore}^\star(\lambda) = \frac{\|\lambda\|_1^2/\|\lambda\|_2^2 - 1}{d - 1}.$$

This immediately recovers the intended boundary cases: if all $d$ eigenvalues are equal, then $\|\lambda\|_1^2/\|\lambda\|_2^2 = d$ and $\text{IsoScore}^\star = 1$; if exactly one eigenvalue is nonzero, then $\|\lambda\|_1^2/\|\lambda\|_2^2 = 1$ and $\text{IsoScore}^\star = 0$.

The same closed form also makes the limitation transparent. IsoScore* depends only on the spectrum through the ratio $\|\lambda\|_1^2/\|\lambda\|_2^2$; it is therefore blind to the eigenspaces carrying that spectrum. Two covariance matrices with the same eigenvalues but different carrier subspaces have exactly the same IsoScore*. For subspace attribution, this means that IsoScore* alone cannot tell whether the dominant anisotropic mass is tangent-aligned, normal-aligned, or distributed across a mixture of both.

A concrete example makes this distinction explicit. Consider an ambient dimension $d = 4$ and a covariance spectrum

$$\lambda = (10^2, 10, 1, 10^{-3}),$$

for which $\text{IsoScore}^\star(\lambda) \approx 0.073$. Yet the spectrum is already highly concentrated: the top eigenvalue alone carries about $90.1\%$ of the variance, and the top two together about $99.1\%$. Suppose now that the tangent proxy coincides with those first two dominant directions. After removing that subspace, the residual positive spectrum becomes

$$\lambda_{\text{res}} = (1, 10^{-3}),$$

which, under the same ambient normalization $d = 4$, yields $\text{IsoScore}^\star(\lambda_{\text{res}}) \approx 0.002$. This is smaller, but not a contradiction: the residual is more anisotropic because one remaining direction still dominates the other and by three orders of magnitude now. Thus, even when the removed tangent subspace clearly carries almost all of the dominant spectral mass, the residual IsoScore* need not by itself identify that carrier subspace.

This is why we do not interpret IsoScore* removals in isolation. In the main paper, IsoScore* is paired with two additional comparisons that are subspace-sensitive: the gradient-energy concentration test, which asks whether $Q_T$ captures unusually

*Figure 4.* Forest Cover Type Anisotropic Components.

large energy relative to matched normal alternatives, and the direct $T > N$ sign test, which asks whether tangent removal improves isotropy more consistently than matched-normal removal across anchors. These comparisons do not replace IsoScore*; they provide the geometric attribution that a global scalar anisotropy statistic cannot provide on its own. In that sense, IsoScore* is best viewed here as a strong compact descriptor of spectral anisotropy, but not as a standalone identifier of the subspace that carries it.

## G. Toy Experiment: Gradient-Concept Alignment

To complement the cross-model true-gradient study in Section 3, we conduct a controlled toy experiment that directly tests the tangent gradient dominance hypothesis (Proposition 2.3) in a simplified setting.

**Motivation: simulating language structure.** The Forest Cover Type dataset provides a tractable proxy for language modeling. Like language data, it features: (i) discrete, categorical inputs (44 binary one-hot features encoding soil type and wilderness area, analogous to token identities); (ii) continuous covariates (10 terrain features, analogous to contextual variation); and (iii) data imbalance, mimicking a stronger biased law than the Zipfian frequency distribution of natural language (See Figure 4). This structure allows us to study anisotropy emergence without the computational cost of full language model training.

**Model and training.** We train a small 2-layer Transformer ($d_{\text{model}} = 56$, $d_{\text{ff}} = 256$, 4 attention heads, $\approx$100k parameters) on Forest Cover Type classification using AdamW with learning rate $10^{-3}$ and batch size 256 for 50 epochs. Throughout training, we extract checkpoints at 1% intervals to track representation dynamics.

**Concept extraction.** At each checkpoint, we apply Semi-NMF, PCA, and ICA to extract $k = 50$ concept directions from: (i) training data $\mathbf{X}$; (ii) intermediate activations $\mathbf{A}^{(\ell)}$; (iii) weight matrices $\mathbf{W}^{(\ell)}$; and (iv) gradient matrices $\nabla_{\mathbf{W}^{(\ell)}} \mathcal{L}$. Concepts are normalized and compared via absolute cosine similarity.

**Alignment metrics.** For concept sets $\mathbf{C}_A$ and $\mathbf{C}_B$, we compute the correlation matrix $R_{ij} = |\hat{\mathbf{c}}_i^A \cdot \hat{\mathbf{c}}_j^B|$ and report: (i) *best-match alignment*: $\bar{r}_{\text{best}} = \frac{1}{k} \sum_j \max_i R_{ij}$; (ii) *top-10 alignment*: mean of the 10 highest best-match correlations; (iii) *weighted alignment*: correlations weighted by concept importance (explained variance for PCA, coefficient norms for Semi-NMF).

**Gradient-data alignment.** The key test of Proposition 2.3 is whether gradient concepts align with data concepts. We track $\bar{r}_{\text{best}}(\mathbf{C}_{\text{grad}}^{(\ell)}, \mathbf{C}_{\text{data}})$ throughout training. The hypothesis predicts high alignment as gradients concentrate along data-derived tangent directions. It is important to note here that we do not compare direct data alignment, but rather data concepts already extracted with concepts present in the gradient updates (seen as a data matrix).

**Key findings.** The toy experiment confirms several predictions: (i) gradient-data concept alignment appears relatively high during training even if it fluctuates. Furthermore, for PCA and semiNMF, it is always highly correlated (See Figure 5); (ii) weight concepts align quickly with activation concepts, indicating that learned parameters do preferably organize around the dominant concepts directions (See Figure 6).

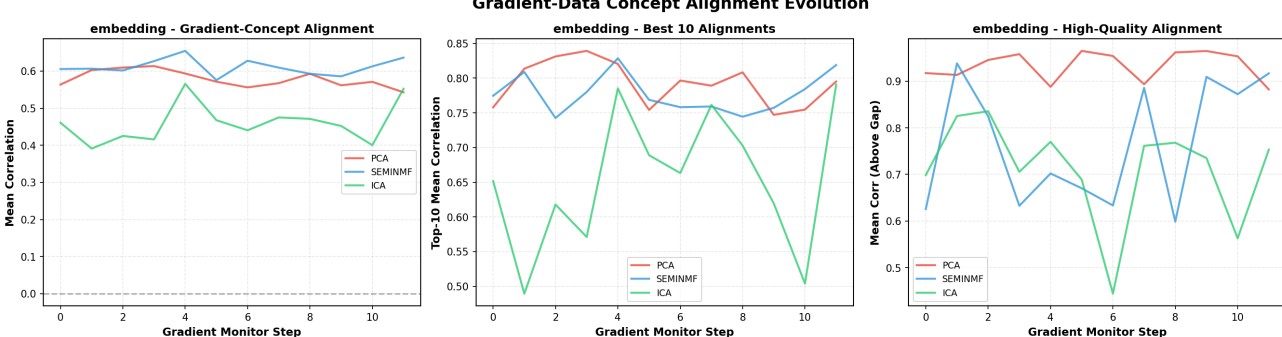

*Figure 5.* Evolution of gradient-data concept alignment for the embedding layer during training. We report three complementary alignment metrics as a function of the gradient monitoring step: (left) mean correlation over all extracted concepts, (middle) mean correlation restricted to the top-10 highest-aligned concepts, and (right) mean correlation for high-quality alignments (above the spectral gap threshold). Data concepts are compared against gradient-derived concepts extracted from the embedding layer using three decomposition methods: PCA (red), Semi-NMF (blue), and ICA (green). Across all metrics, PCA and Semi-NMF exhibit consistently stronger and more stable alignment than ICA, with Semi-NMF showing particularly strong alignments.

## H. Experimental Setup Details

### H.1. Main Experiments

The code used for this experiment is available here: https://github.com/Raphael-Bernas/Revisiting-Anisotropy-In-Language-Transformers.git.

The primary experiments of Section 3 are run on ten language models spanning encoder-style and decoder-style architectures: EuroBERT-210m, EuroBERT-610m, moderncamembert-base, OLMo-1B, pythia-160m, pythia-410m, pythia-1b, Gaperon-1.5B, SmolLM2-360M, and SmolLM2-1.7B. For every model we use the same merged dataset `revisited_mix`, obtained by round-robin sampling from `allenai/c4` (English), `Salesforce/wikitext` with `wikitext-103-raw-v1`, `HuggingFaceFW/fineweb-edu` with `sample-10BT`, `ccdv/arxiv-summarization` using the article field, and `wikimedia/wikipedia` with `20231101.fr`. The resulting corpus contains 6000 sequences of length 128, with 1200 sequences from each source.

Anchor tokens are selected after special-token filtering and a minimum occurrence threshold, then stratified into 4 frequency bins with 6 anchors per bin, for a total of 24 anchors per model. Each anchor receives 24 fit contexts and 8 evaluation contexts sampled without fit/eval overlap. Early and late phases correspond respectively to the first 30% and last 30% of the available training trajectory, subsampled uniformly to at most 6 and 4 checkpoints. Tangent fits use PCA with 90% explained-variance target, minimum rank 4, maximum rank 12, and the main paper reports the early concatenation context regime in which the early activation-derived basis is fit once and reused throughout evaluation.

True gradients are computed by ordinary backpropagation of the model loss on the evaluation contexts (decoder and encoder adapted). The reported energy null uses 20 matched-rank random normal samples, and the anisotropy statistics use a shrinkage-regularized IsoScore* computation with $\alpha = 0.05$. The paper-facing summaries first average checkpoint-level quantities within each anchor and then aggregate across anchors, which is why the main table emphasizes effect sizes together with explicit Monte Carlo and sign-test p-values rather than standard deviations.

### H.2. Complementary Experiments

The supplementary mechanistic-interpretability analysis in Appendix D uses EuroBERT-210M, Pythia-410M, and SmolLM2-360M checkpoints, a fixed `Salesforce/wikitext` subset, and Semi-NMF concept extraction with 600 components. It reports intrinsic-dimension, entropy, spectral, and weight-alignment statistics as complementary descriptive evidence about how activation geometry evolves during training. Appendix E adds a second supplementary view based on realized embedding updates $dW$, reporting tangent and normal enrichments relative to random across token-frequency groups and representative tracked tokens.

Our experiments were conducted on three transformer language models of varying sizes: EuroBERT-210m (48 intermediate checkpoints), Pythia-410m (32 intermediate checkpoints), and SmolLM2-360M (16 intermediate checkpoints). All models

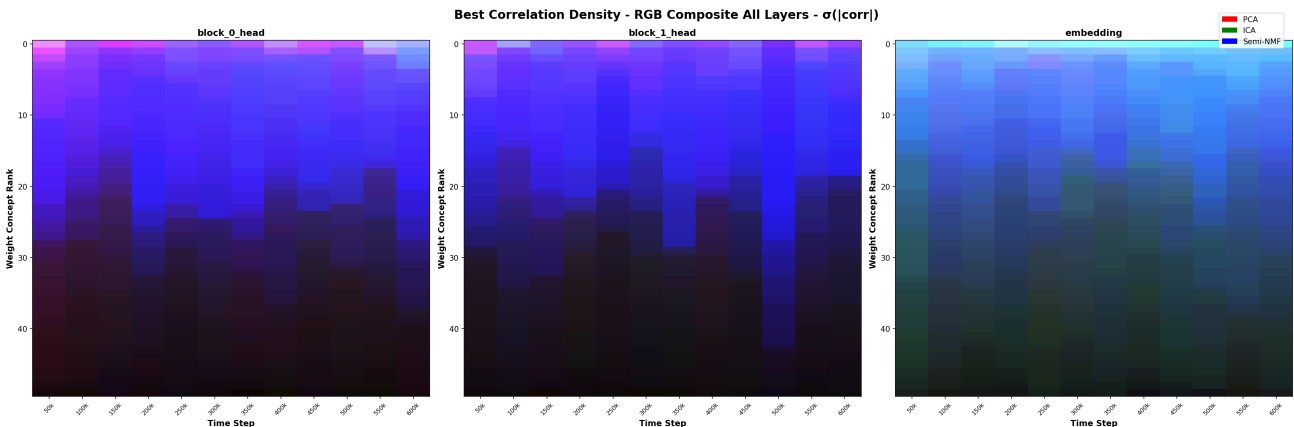

*Figure 6.* Correlation density maps for the three main components of the model: the first transformer linear head, the second transformer linear head, and the embedding matrix (first layer), shown as a function of training time (x-axis). To construct these maps, we first pass data-derived concepts through the model to obtain an estimate of the concepts learned by the network. Independently, we treat the weight matrices of the corresponding linear layers as data matrices and extract latent concepts using PCA, ICA, and Semi-NMF. The resulting concept weights from each decomposition are used to rank the extracted components (rank 0 corresponds to the highest-weight concept, rank 50 to the lowest). For each rank, we compute the absolute correlation between learned concepts and data concepts and visualize these values as an RGB image, with PCA encoded in red, ICA in green, and Semi-NMF in blue. Brighter colors indicate stronger correlations. Across all layers, higher-ranked concepts consistently exhibit stronger correlations throughout training.

were analyzed using the Salesforce/wikitext dataset (wikitext-103-raw-v1 split), with 100 samples per checkpoint, each truncated to approximately 100 tokens, with fixed seed.

For concept extraction, we employed Semi-Nonnegative Matrix Factorization (SemiNMF) with 600 concepts components (a value under hidden dimension is mandatory) using an Interpreto[4] (Poché et al., 2025) / Overcomplete[5] backend. Activations were pooled using mean pooling across the sequence dimension. Our analysis pipeline computed multiple metrics including: (i) intrinsic dimension estimates via two-Nearest Neighbors (twoNN) and graph-based methods, (ii) entropy measures from graph analysis, (iii) anisotropy metrics including spectral gap, $PCA_{70\%}$, and eigenvector stability, (iv) effective rank of concept representations, and (v) weight correlation analysis between consecutive layers. All intermediate checkpoints were processed to capture the full training dynamics across model development.

---

[4]https://github.com/FOR-sight-ai/interpreto.git

[5]https://github.com/KempnerInstitute/overcomplete.git

