# OpenReview forum: "Revisiting Anisotropy in Language Transformers: The Geometry of Learning Dynamics"
_ICML.cc/2026/Conference — ICML 2026 regular_

### Official Review · Reviewer_mxbz · 2026-03-12

**Soundness:** 3
**Presentation:** 3
**Significance:** 3
**Originality:** 3
**Overall Recommendation:** 4
**Confidence:** 3

**Summary:**

This paper studies, both theoretically and empirically, anisotropy in language model embeddings. In particular, the authors argue that training dynamics favor the tangent-space of the manifold, resulting in representations that concentrate on a low-curvature, low-dimensional structure. They first show that the geometric curvature bias results in high-curvature regions of the manifold becoming under-represented in the marginal distribution, suppressing curvature-derived features. They then show that gradient updates reinforce tangent directions over normal directions, which further suppresses these same curvature-derived features. Following this theoretical analysis, they present small-scale experiments to test whether gradient updates preferentially reinforce tangent directions, producing a low-dimensional, approximately linear subspace that dominates the representation geometry, by applying Semi-NMF to the activations and gradients, indeed finding gradient alignment with the discovered concept space.

**Compliance With Llm Reviewing Policy:**

Affirmed.

**Key Questions For Authors:**

1. Figure 5 is very difficult to understand, is there another way you can plot the same data?
2. What are the 'Gradient Monitor Steps' in Figure 4?

**Limitations:**

The authors clearly state the limitations of their theoretical analysis and does not pose any direct negative societal impact, as stated by the authors.

**Strengths And Weaknesses:**

Strengths:
- This paper addresses a very interesting phenomenon that has particularly relevant implications for the interpretability of learning dynamics.
- The theory is intuitive and the narrative is easy to follow.
- The empirical results validate the theoretical claims made in the paper, and again are easy to understand.
- I am relatively unfamiliar with the recent literature on anisotropy in LLMs, but the links drawn by this paper between learning dynamics and the geometry of representations in LLMs seem quite novel to me and would be of interest to the community more broadly.

Weaknesses:
- Can you expand further on the claim made in the conclusion: "In fact, by concentrating representations into lowdimensional structure, it may effectively move overparameterized Transformers toward an underparameterized regime that favors generalization." This seems like a very important claim but is not discussed directly in the main paper.
- The analysis of the results in Table 1 is very limited, and the paper would be significantly improved by more thoroughly discussing the results, in particular the discrepancies between the different models, and the lack of change over training steps (e.g. most values are relatively stable from left to right). I would have expected much stronger temporal dynamics than what is shown here.

I am willing to improve my score if the authors address these weaknesses/questions.

---

> ### Author Rebuttal · Authors · 2026-03-31
>
> We thank the reviewer for the positive assessment and for the very constructive suggestions. We are especially encouraged by your comment that the link between learning dynamics and representation geometry may be of broader interest. We agree that the current version can do a better job on three points: (i) clarifying the sentence on anisotropy and generalization, (ii) expanding the interpretation of Table 1, and (iii) improving the presentation of Figures 4-5.
>
> First, regarding the conclusion claim that anisotropy may move overparameterized Transformers toward a more underparameterized regime that favors generalization: we agree that this sentence was too compressed relative to the discussion in the main text. This is not the main claim of the paper. Our intended point is more modest: if training progressively concentrates representation energy and update dynamics into a smaller effective subspace, then anisotropy may act as an implicit structural bias compatible with improved generalization. This intuition is motivated by observations from low-dimensional adaptation and fine-tuning, and, as discussed in our response to Reviewer Dk1w, it is also consistent with recent anisotropy literature showing that anisotropic representations can be beneficial. We will therefore revise this sentence to make clear that it is an interpretation and motivation, while the main contribution of the paper is to explain why anisotropy may arise naturally during Transformer training, and in particular why tangent directions may be preferentially amplified.
>
> Second, we agree that Table 1 deserves a substantially richer discussion. The main signal we intended the table to convey is not only raw temporal drift from left to right, but also the persistent geometric separation between different kinds of quantities: intrinsic/support-like measures remain much smaller than linear spectral quantities, while alignment- and entropy-based quantities track the stabilization of the dominant representation geometry. We also agree that the differences between models should have been discussed more explicitly. In particular, one of the three models is an encoder while the other two are decoders, and the decoder trends are in fact much more similar to each other than to the encoder. The apparently limited temporal variation is also partly explained by checkpoint granularity. For several public models, the first available checkpoints already correspond to a relatively advanced training stage, so the earliest and strongest internal reorganization is only partially visible in the released trajectories. In the broader analyses we have done so far, we observe that much of the geometric organization forms early and then stabilizes, after which training continues with weaker variation in these global summaries. This is precisely one reason we included the entropy-based metric: it acts as a proxy for whether internal organization is still evolving even when several geometric statistics appear visually stable. We will further clarify these interpretation in the revised paper.
>
> Regarding your specific questions: the "Gradient Monitor Steps'' in Figure 4 denote the indices of the monitored gradient checkpoints. Concretely, we save gradients every 5 epochs and, at each monitored step, decompose the gradient and evaluate how strongly it aligns with the tangent proxy. In this setting, a strong correlation means that the gradient carries disproportionately more tangent-direction signal than non-tangent signal. We will clarify this explicitly in the caption and main text.
>
> For Figure 5, we agree that the current visualization is harder to read than it should be. The purpose of that figure is to show how concept directions extracted from data activations-used here as proxies for the dominant shared directions in the data, and therefore for the tangent structure emphasized by training-align with internal model weights. Intuitively, it is meant to visualize to what extent the model's weights organize themselves around the dominant directions discovered from the data. We will replace the figure with a clearer visualization and expand the accompanying explanation.
>
> Overall, we view your comments as mainly calling for clearer interpretation and presentation rather than a change to the central thesis, and we appreciate that. We believe the revised discussion of Table 1, the clarification of the conclusion, and the new experiments added in response to the other reviews will make the paper substantially stronger.

---

> > ### Author Rebuttal · Reviewer_mxbz · 2026-04-05
> >
> > The authors directly addressed my questions and comments, although I believe that my first weakness comment still warrants either a more extended literature review on the topic or empirical evidence to back up the claims. For this reason, I will maintain my current score of weak accept.

---

> > > ### Author Response · Authors · 2026-04-05
> > >
> > > We thank the reviewer for the positive follow-up and for indicating that the main concerns have largely been resolved. On the remaining reservation regarding the first weakness comment, we believe the clearest way to address it is to distinguish the paper’s main theoretical claim from a broader interpretive remark in the conclusion.
> > >
> > > Improvement of generalization through anisotropy is not our paper's claim. Our central claim is the tangent-alignment mechanism: frequency-biased training preferentially amplifies tangent directions and attenuates normal directions, providing a geometric explanation for how anisotropy can emerge during Transformer training. This is the contribution stated in the introduction. The sentence in the conclusion was intended as an interpretation and motivation for future work. But we agree that this interpretation is better grounded in the recent literature already motivating this idea. Under reviewer Dk1w’s suggestion, we revised the introduction and related-work sections so that it is now explicitly tied to recent literature. In particular, Rudman et al report that increasing anisotropy often improves downstream performance and explicitly notes that this is aligned with earlier work on anisotropy and generalization [1]. That earlier work shows that anisotropic noise helps optimization escape sharp minima and move toward flatter minima that typically generalize better [2]. In parallel, Mickus et al show that strict isotropy is not always compatible with linearly useful representations [3]. Thus, we now present the generalization-oriented sentence as a literature-motivated interpretation. And our paper is clearly aligned with this broader perspective since the theoretical reasoning suggests that, during training, internal representations become increasingly aligned with tangent directions, while normal directions - those that reflect curvature and more subtle local variation - are progressively attenuated. Geometrically, this means that the representation dynamics become concentrated in a lower-dimensional subspace. Intuitively, such a reduction in expressivity is compatible with the idea that structured anisotropy may favor generalization. But again, this is better viewed as an interpretation of the mechanism, already motivated by the literature above, rather than as a claim our paper seeks to establish.
> > >
> > > Independently of that, we have also strengthened the empirical support for the paper’s actual theoretical claim. As requested by reviewers Dk1w and 9chP, we now report new statistical tests and quantitative results over **5 datasets** and **8 models** (**5 decoders, 3 encoders**). Reviewer Dk1w suggested adding IsoScore\*, which is a strong global anisotropy metric: it quantifies how uneven the covariance spectrum is. This is useful, but it does not identify **which subspace** carries that anisotropy (\*). We therefore complement it with our tangent/normal enrichment metric, reported as `E-r` in the response to reviewer 9chP, which measures how much more the true gradient $\nabla L$ aligns with the tangent space than with matched normal alternatives. Those tables show a strong and consistent signal: tangent-aligned directions explain substantially more gradient energy than matched-normal controls, often by about one order of magnitude and sometimes much more, while normal-direction enrichment remains near or below the random baseline. In our response to reviewer Dk1w, we additionally report a complementary update-level analysis on the actual parameter evolution, using $dW$ rather than $\nabla L$, together with tracked-token frequencies. This provides a broader picture of the same mechanism, and shows that for frequent tokens tangent enrichment is typically at least one order of magnitude larger.
> > >
> > > For these reasons, we believe the remaining concern is now addressed on both fronts: the broader interpretive sentence is better grounded in the revised literature review, and the paper’s main theoretical claim is now supported by substantially stronger direct experiments.
> > >
> > > ### References
> > >
> > > [1] *Stable Anisotropic Regularization*. Rudman et al. 2024.
> > >
> > > [2] *The Anisotropic Noise in Stochastic Gradient Descent: Its Behavior of Escaping from Sharp Minima and Regularization Effects*. Zhu et al. 2018.
> > >
> > > [3] *Isotropy, Clusters, and Classifiers*. Mickus et al. 2024.
> > >
> > > (\*) IsoScore* is very informative about the amount of spectral anisotropy, but not by itself about its geometric carrier. For example, the spectrum $(10^2,10,1,10^{-3})$ has IsoScore* $\approx 0.073$: the top eigenvalue explains about $90.1%$ of the variance, and the top two about $99.1%$. If we remove those first two tangent-associated directions, the residual spectrum becomes $(1,10^{-3})$, whose IsoScore* is even lower, $\approx 0.002$. This does not contradict the metric: it remains highly sensitive to residual spectral imbalance, but it does not by itself identify which subspace originally carried the dominant anisotropic mass.

---

### Official Review · Reviewer_Dk1w · 2026-03-13

**Soundness:** 2
**Presentation:** 2
**Significance:** 2
**Originality:** 2
**Overall Recommendation:** 2
**Confidence:** 4

**Summary:**

This paper revisits the anisotropy phenomenon in Transformer language models from a geometric perspective. The authors argue, via a differential-geometric framework built on a data manifold hypothesis, that the syntactic geometry of language favors a linear subspace, and that frequency-biased sampling suppresses the visibility of high-curvature directions on the embedding manifold. They connect this geometric bias to training dynamics, arguing that gradient updates preferentially reinforce tangent directions, creating a self-amplifying mechanism that produces and sustains anisotropy. To observe these dynamics empirically, they apply mechanistic interpretability techniques and local geometric metrics (TwoNN intrinsic dimension, spectral entropy, effective rank, concept direction alignment) at training checkpoints of EuroBERT, Pythia 410M, and SmolLM2. Their central empirical claim that anisotropy is not fundamentally detrimental to models and may hep with concentrating models into a lower effective parameter space.

**Compliance With Llm Reviewing Policy:**

Affirmed.

**Key Questions For Authors:**

- Given that Rudman and Eickhoff (2024) demonstrated that increasing anisotropy improves downstream performance, and that Mickus et al. (2024) proved mathematically and empirically that isotropy is structurally incompatible with clustering and linear classification objectives, how do the authors reconcile their framing of anisotropy as a significant challenge? What measure of isotropy do the authors consider valid, and why is it not used throughout the paper?

**Limitations:**

yes

**Strengths And Weaknesses:**

__Strengths__

- The paper takes an interesting angle by applying mechanistic interpretability techniques to training checkpoints rather than post-hoc, which is a meaningful methodological departure from most prior work in this space. The geometric framing, specifically the distinction between intrinsic geodesic distances and ambient Euclidean chords, is a thoughtful conceptual contribution that adds nuance to how the community thinks about representation geometry. The analysis spanning encoder and decoder architectures gives the empirical component some breadth. The theoretical argument linking frequency-biased sampling to tangent-space amplification is creative.

__Weaknesses__

- The paper's treatment of the isotropy/anisotropy literature is a serious problem. The framing in the introduction and theoretical sections implicitly assumes anisotropy is a deficiency to be explained and remediated. This is not a defensible starting position given recent results. Rudman and Eickhoff (2024, I-STAR) demonstrated with a properly calibrated isotropy measure that decreasing isotropy during training actually improves downstream performance across the majority of tasks and models evaluated [1] . Critically, Mickus et al. (2024) provide a formal mathematical and empirical result that is directly relevant here [2]. They demonstrate that strict isotropy, as measured by IsoScore (Rudman et al., 2022), is structurally incompatible with the presence of clusters in the embedding space, and that any optimum for a linear classification objective necessarily violates the isotropy requirement [3]. They verify this both mathematically and empirically on real-world datasets using silhouette scores as a clustering metric, showing that enforcing isotropy actively degrades classification performance. The reviewed paper does not engage with this body of work at all, which fundamentally undermines the premise that anisotropy is a problem requiring geometric explanation.

- The paper's definition and operationalization of anisotropy are outdated. Anisotropy is properly defined as the non-uniformity of variance across embedding dimensions, and prior work has rigorously established that average cosine similarity is not a valid measure of this property [3]. The paper continues to lean on cosine-based intuitions and geometric framings without adequately addressing this critique.

- Semi-NMF applied to pooled activations is not a mechanistic interpretability technique in any standard sense of the term. Describing concept direction extraction via matrix factorization as MI is a significant overstatement and will mislead readers about the methodological contribution.

- Table 1 is nearly unreadable and is also the primary empirical result of the paper. Without a legible presentation of the evidence, the theoretical claims are hard to evaluate.

[1] Stable Anisotropic Regularization. Rudman et al. 2024
[2] Isotropy, Clusters and Classifiers. Mickus et al. 2024
[3] IsoScore: Measuring the Uniformity of Embedding Space Utilization. Rudman et al. 2022

---

> ### Author Rebuttal · Authors · 2026-03-31
>
> We thank the reviewer for the careful reading and for pointing us to relevant recent work on isotropy and anisotropy. We agree these references should have been discussed explicitly. They are also aligned with our intended position: our paper is not centered on whether anisotropy should be promoted or removed, but on the geometric mechanism by which anisotropic structure emerges during training.
>
> Our introduction already states that anisotropy should not be reduced to a purely negative artifact, and our conclusion makes the same point explicitly: the paper studies how frequency-biased training can induce directional concentration in representation space. In this sense, the works cited by the reviewer are complementary to our perspective. Stable Anisotropic Regularization shows that decreasing isotropy can improve downstream performance under a stronger isotropy metric [1], while Mickus et al. clarify that strict isotropy is structurally incompatible with clustered and linearly useful representations [2,3]. We agree the manuscript should have made this alignment more explicit, and we will revise the introduction and related work accordingly.
>
> We also appreciate the reviewer’s point about isotropy measurement. We do not view average cosine similarity as a sufficient measure of isotropy, and the empirical picture in the paper is not built on cosine quantities alone: it also uses local intrinsic-dimension estimates, spectral entropy, and effective-rank diagnostics across checkpoints. Thus, the core claims do not rest purely on cosine similarity. That said, IsoScore was introduced precisely because earlier cosine-style proxies are not reliable as global isotropy measures [3], and we will make this distinction clearer in the revision.
>
> On the empirical side, we have already extended the study with a more targeted tangent-bias experiment. For each tracked token, we estimate a local tangent space and test whether checkpoint updates are preferentially concentrated in that space relative to dimension-matched random subspaces, using a dimension-normalized overlap statistic. Across all past and newly added models, we observe the expected trend: for frequent tokens, tangent-space enrichment is consistently much stronger than for rare tokens, typically around **10-50$\times$** larger, while rare-token enrichments are often close to random. At the same time, normal-space enrichments remain near random in many cases, and for frequent tokens are often slightly below the random baseline, whereas tangent enrichments remain at least **10$\times$** stronger than random. We believe this directly strengthens the paper’s central geometric claim.
>
> This revised pipeline also makes it straightforward to add an IsoScore$^\star$-based test, in the spirit of reviewer Dk1w’s suggestion, by evaluating isotropy before and after removing the tangent component and comparing against matched null subspaces with Monte Carlo estimation. We are currently integrating this statistic into the same framework. We have also expanded the study beyond the original three models by adding **OLMo-1B**, **ModernCamemBERT**, and larger variants from the same families when checkpoint availability permits.
>
> Regarding mechanistic interpretability, we agree the scope should be stated more precisely, but we do not think the positioning is misleading. As discussed by Saphra and Wiegreffe [4], *mechanistic interpretability* is used in multiple senses, including a broader technical sense covering analyses of internal representations and computations even without a fully specified causal circuit. Our work studies internal representations and update geometry across training to explain a global representational phenomenon in terms of internal structure, which fits this broader use. Likewise, [5] and [6] explicitly use Semi-NMF-style decomposition of pooled activations as a tool for concept-oriented interpretability; in particular, [6] includes Semi-NMF within dictionary-learning approaches for concept extraction. We are happy to clarify that our use of Semi-NMF is one component of a broader analysis rather than the sole defining element of mechanistic interpretability.
>
> Finally, we agree that Table 1 should be improved. In the revision, we will present the main empirical evidence more clearly and incorporate the extended tangent-bias analysis above. Overall, the cited works help sharpen the manuscript’s framing, while the new experiments provide a more direct empirical test of the geometric mechanism studied in the paper.
>
> ### References
> Only in this response, we reuse the reviewer’s numbering for [1], [2], and [3].
>
> [4] Saphra, N., & Wiegreffe, S. (2024). *Mechanistic?*
>
> [5] Shafran, O., Geiger, A., & Geva, M. (2025). *Decomposing MLP Activations into Interpretable Features via Semi-Nonnegative Matrix Factorization*.
>
> [6] Fel, T., et al. (2025). *Archetypal SAE: Adaptive and Stable Dictionary Learning for Concept Extraction in Large Vision Models*.

---

> > ### Author Rebuttal · Reviewer_Dk1w · 2026-04-01
> >
> > Thank you for the detailed response. Although the authors say they will improve results before revision, I would need to see full, improved results before changing my score.

---

> > > ### Author Response · Authors · 2026-04-05
> > >
> > > We thank the reviewer for the follow-up. As requested, we provide new experiments. The full IsoScore* study on backpropagated gradients is reported in our response to reviewer **9chP**. Since those gradients are computed on our evaluation batches rather than on the one used at each original pretraining step, we carried out a complementary update-level analysis on the actual parameter evolution by measuring $dW = W_{t+n} - W_t$ on the embedding matrix, where $n$ is the available checkpoint granularity. We view the true-gradient study as the cleanest validation of the theory; the $B=dW$ study below is complementary and shows that the same phenomenon is visible in the effective update. Using same notations as in our response to reviewer 9chP, we report tangent and normal enrichments $\mathcal E(Q_T)$ and $\mathcal E(Q_N)$ computed on $B=dW$, with the random baseline normalized to $1$. The signal is strong and highly consistent. Tangent enrichment is systematically larger than normal enrichment, especially for frequent tokens. In both the gradient-based and update-based experiments, the effect weakens from the earlier to the later available checkpoints as suggested in the paper, but much of the representational organization forms early, and later training proceeds with a substantially smaller gradient signal, so anisotropy is already present and the remaining drift is weaker. Importantly, our “early” segment already lies in roughly the first half of pretraining, not near initialization; even in this already relatively late regime, the tangent-bias remains clear. The table below reports representative tracked tokens together with their frequencies and early/stable enrichments. Across models, frequent tokens typically show strong tangent enrichment, often one to two orders of magnitude above the random baseline, whereas normal enrichment stays at or slightly below random. This is consistent with the geometric mechanism proposed in the paper.
> > >
> > > | Model | Token | Freq. | Early Tangent | Early Normal | Late Tangent | Late Normal |
> > > |---|---|---:|---:|---:|---:|---:|
> > > | **EuroBERT-210m** | `of` | 2.40e-02 | 33.27 | 0.99 | 3.72 | 1.00 |
> > > |  | `her` | 1.63e-03 | 7.40 | 1.00 | 3.53 | 1.00 |
> > > |  | `lines` | 1.16e-04 | 2.79 | 1.00 | 2.24 | 1.00 |
> > > |  | `jam` | 6.45e-06 | 3.26 | 1.00 | 2.76 | 1.00 |
> > > |  | `chat` | 2.15e-06 | 3.63 | 1.00 | 2.02 | 1.00 |
> > > | **EuroBERT-610m** | `the` | 4.58e-02 | 28.24 | 1.00 | 1.94 | 1.00 |
> > > |  | `after` | 1.11e-03 | 6.66 | 1.00 | 1.48 | 1.00 |
> > > |  | `Christmas` | 1.38e-04 | 4.02 | 1.00 | 1.36 | 1.00 |
> > > |  | `streaming` | 2.15e-06 | 6.04 | 1.00 | 1.57 | 1.00 |
> > > |  | `chat` | 2.15e-06 | 7.16 | 1.00 | 1.37 | 1.00 |
> > > | **ModernCamemBERT** | `##s` | 1.48e-02 | 53.90 | 0.99 | 22.16 | 0.98 |
> > > |  | `his` | 2.46e-03 | 4.07 | 1.00 | 1.90 | 1.00 |
> > > |  | `corn` | 1.02e-04 | 3.10 | 1.00 | 2.05 | 1.00 |
> > > |  | `lib` | 2.41e-05 | 4.41 | 1.00 | 1.75 | 1.00 |
> > > |  | `##acha` | 2.01e-06 | 3.98 | 1.00 | 2.18 | 1.00 |
> > > | **OLMo-1b** | `for` | 5.62e-03 | 16.23 | 1.00 | 5.45 | 0.96 |
> > > |  | `but` | 1.31e-03 | 4.88 | 1.00 | 1.94 | 0.99 |
> > > |  | `love` | 1.49e-04 | 7.17 | 1.00 | 1.90 | 0.99 |
> > > |  | `helping` | 1.95e-05 | 13.61 | 1.00 | 1.96 | 0.99 |
> > > |  | `robust` | 4.33e-06 | 14.93 | 1.00 | 2.64 | 0.99 |
> > > | **Pythia-1b** | `to` | 1.58e-02 | 128.97 | 0.99 | 15.80 | 0.99 |
> > > |  | `into` | 9.57e-04 | 3.66 | 1.00 | 6.88 | 1.00 |
> > > |  | `Canadian` | 1.78e-04 | 3.01 | 1.00 | 5.29 | 1.00 |
> > > |  | `gains` | 1.08e-05 | 2.58 | 1.00 | 6.51 | 1.00 |
> > > |  | `generator` | 2.16e-06 | 3.27 | 1.00 | 5.45 | 1.00 |
> > > | **Pythia-160m** | `the` | 4.65e-02 | 58.52 | 0.98 | 1.99 | 0.98 |
> > > |  | `also` | 1.46e-03 | 4.08 | 1.00 | 2.04 | 0.99 |
> > > |  | `poem` | 1.21e-04 | 2.05 | 1.00 | 4.47 | 0.98 |
> > > |  | `gains` | 1.08e-05 | 1.77 | 1.00 | 3.26 | 0.98 |
> > > |  | `Dual` | 2.16e-06 | 1.89 | 1.00 | 3.66 | 0.99 |
> > > | **Pythia-410m** | `to` | 1.58e-02 | 27.85 | 1.00 | 14.59 | 0.99 |
> > > |  | `It` | 1.05e-03 | 2.92 | 1.00 | 4.89 | 1.00 |
> > > |  | `signed` | 1.43e-04 | 2.32 | 1.00 | 4.75 | 1.00 |
> > > |  | `instinct` | 8.66e-06 | 2.06 | 1.00 | 5.54 | 1.00 |
> > > |  | `doctrine` | 4.33e-06 | 2.55 | 1.00 | 4.44 | 1.00 |
> > > | **SmolLM2-1.7b** | `c` | 2.22e-02 | 112.84 | 0.97 | 23.42 | 0.96 |
> > > |  | `M` | 2.49e-03 | 41.13 | 0.99 | 10.65 | 0.98 |
> > > |  | `L` | 1.64e-03 | 37.79 | 0.99 | 10.20 | 0.97 |
> > > |  | `x` | 1.33e-03 | 51.25 | 0.98 | 7.79 | 0.96 |
> > > |  | `q` | 5.32e-04 | 82.81 | 0.98 | 9.82 | 0.95 |
> > > | **SmolLM2-360m** (*) | `i` | 5.29e-02 | - | - | 7.29 | 1.00 |
> > > |  | `y` | 1.10e-02 | - | - | 4.12 | 0.99 |
> > > |  | `x` | 1.33e-03 | - | - | 4.13 | 0.99 |
> > > |  | `q` | 5.32e-04 | - | - | 5.28 | 0.99 |
> > > |  | `X` | 6.73e-05 | - | - | 6.26 | 0.99 |
> > >
> > > This reinforce the theory: Updates are more aligned with tangent than normal directions; with strongest contrast for frequent tokens; normal enrichment staying slightly below random. Although the effect weakens at later checkpoints, it remains clearly present throughout the available training trajectory.
> > > (*) SmolLM2-360m only has 16 available checkpoints starting in the late training phase.

---

### Official Review · Reviewer_9chP · 2026-03-13

**Soundness:** 3
**Presentation:** 2
**Significance:** 2
**Originality:** 3
**Overall Recommendation:** 3
**Confidence:** 3

**Summary:**

This paper mainly studies the underlying mechanisms of anisotropy in Transformer language models from the perspective of differential geometry and training dynamics. The paper's main contribution lies in connecting the geometric biases to Transformer learning dynamics, which causes model representation to gradually exhibit anisotropy and shrink towards a lower-dimensional subspace during training. By tracking the model's training dynamics and introducing interpretable tools, this paper interprets the evolution of the model's representation space.

**Compliance With Llm Reviewing Policy:**

Affirmed.

**Final Justification:**

Thank the authors for their response. Given the lack of substantial data supporting the original text, and the author's inability to fully supplement the data to support the article's arguments during the rebuttal phase, I will maintain my score.

**Key Questions For Authors:**

- Can the authors provide a statistical measurement of the gradient norm during model training?
- Can the authors further validate the theory on more complex datasets, as the current dataset is too simplistic?

**Limitations:**

Yes

**Strengths And Weaknesses:**

## Strengths
- This paper connects geometry with model optimization, making a contribution to the well-known problem of anisotropy in large language models.
- The derivations in the paper are insightful, and through in-depth analysis, they effectively explain the mechanistic changes in the model's evolution process. This theoretical contribution has foundational significance for future work.
## Weaknesses
- The paper's derivation is mainly based on idealized assumptions, which makes it difficult to directly explain the problems of large language models and carries the risk of exaggeration. This has a significant impact on the future impact and practical significance of the work.
- The experiments designed in this paper fail to provide sufficient support for the theory. Some of the statistics (like curvature) mentioned in the paper lack direct statistical observations. The sample size is small, and no experiments with different random number seeds were conducted to verify the consistency of the conclusions. The analysis of most experimental phenomena is overly subjective and lacks data support.

In summary, the theoretical part of the paper needs more extensive experimental verification; otherwise, its true significance cannot be confirmed.

---

> ### Author Rebuttal · Authors · 2026-03-31
>
> We thank the reviewer for the careful reading, and especially for highlighting the potential foundational significance of the paper’s theoretical contribution.
>
> We believe the main clarification concerns the scope and role of the theory. Transformer training dynamics are difficult to interpret without an explicit modeling framework, and our goal is to provide such a framework at the level needed to derive concrete geometric signatures. The analysis is therefore **purely local**: around an idealized reference embedding, we assume that token representations live **near a smooth local patch** of the ambient space. This is exactly the level of regularity required for the tangent/normal decomposition used in the paper. In that sense, (A1)–(A2) are standard local regularity assumptions for differential-geometric analysis. They are also weaker in scope than the geometric picture suggested by Robinson et al. [3], who argue that token spaces are better described as **stratified** rather than globally manifold. Our argument only requires a local regular stratum around $\mu$. The sampling-bias assumption then encodes the empirically observed fact that frequent tokens dominate training and probe a narrower local geometry [1,2]. Seen this way, the theory is a local mechanism model linking frequency bias to directional bias: frequent-token updates preferentially amplify tangent directions while comparatively attenuating locally normal ones, yielding anisotropic concentration during training. This is meant to bridge existing empirical observations on frequency bias and anisotropy [1,2] with a more explicit geometric account of why the effect should arise.
>
> We also appreciate the suggestion to strengthen the empirical side with more direct statistical tests. The original experiments were meant to test whether the signatures predicted by this mechanism already appear in real training trajectories. Because the true tangent space is latent when only public checkpoints are available, we used concept directions extracted from pooled activations as a coarse surrogate. Even with this indirect proxy, the evidence was coherent across models: intrinsic dimension, graph-based dimension, effective rank, entropy, principal-direction stability, and concept/weight alignment all showed behavior consistent with the same anisotropic tangent-flattening picture. We agree that stronger empirical tests make this interpretation considerably clearer.
>
> Following the reviewer’s request for more direct statistics, and inspired also by reviewer **Dk1w**’s suggestion, we designed new experiments specifically targeting this point. First, we now directly measure update/gradient overlap with tangent and normal spaces through a dimension-normalized norm-based statistic. Second, we compare these overlaps against matched random subspaces and estimate significance with Monte Carlo null tests. The signal is strong and consistent across all tested models: for **frequent tokens**, tangent-space enrichment is far above random, with weakest cases still around **10x** the random baseline and strongest close to **100x**, while normal directions remain near random or slightly below it (worst cases around **0.97** when the random baseline is normalized to **1**). This gives a direct norm-based measurement of how training updates concentrate in tangent directions. We also broadened the validation beyond the original Wikitext setup by adding four additional datasets: **allenai/c4**, **HuggingFaceFW/fineweb-edu**, **ccdv/arxiv-summarization**, and **wikimedia/wikipedia**. In addition, we extended the model set with larger variants from the original families, plus **OLMo-1B** and **ModernCamemBERT**.
>
> We believe these additions address the applicability concern directly: the theory remains local and modest in scope, while the revised experiments now test its predicted signatures with direct overlap statistics, broader datasets, and more realistic model settings. This substantially strengthens the connection between the proposed geometric mechanism and observed Transformer training dynamics.
>
> ### References
>
> [1] Gao, J., He, D., Tan, X., Qin, T., Wang, L., & Liu, T.-Y. *Representation Degeneration Problem in Training Natural Language Generation Models*.
>
> [2] Diehl Martinez, R., Goriely, Z., Caines, A., Buttery, P., & Beinborn, L. *Mitigating Frequency Bias and Anisotropy in Language Model Pre-Training with Syntactic Smoothing*.
>
> [3] Robinson, M., Dey, S., & Sweet, S. *The Structure of the Token Space for Large Language Models*.

---

> > ### Author Rebuttal · Reviewer_9chP · 2026-04-02
> >
> > The authors' rebuttal resolved some of my questions about the assumptions, but they did not provide more detailed experimental data to offer better evidence, only a vague description. Therefore, I will maintain my recommendation.

---

> > > ### Author Response · Authors · 2026-04-05
> > >
> > > We thank the reviewer for the follow-up. In our new experiments, we fit a low-rank orthonormal basis $Q_T$ on centered early activations and keep it fixed for early/late evaluation. True gradients $B=\nabla L$ are obtained by standard backpropagation over a mix of the new datasets already discussed. We report the first, middle, and last transformer blocks; for Pythia, `b1_in` is the first block. We provide those metrics for early/late training phase:
> > > - `E-r`: tangent energy density divided by matched-rank normal energy density, (the theory's gradient-norm), $R_E=\mathcal E(Q_T)/\mathcal E(Q_{N,\det}),\ \mathcal E(Q)=||BQ||_F^2/\dim(Q).$
> > > - `E-p`: Monte Carlo p-value against random same-rank normal subspaces in $T^\perp$; with $S=20$, the minimum is $1/21 \approx 0.0476$.
> > > - `ΔI*%(T/N)`: IsoScore* percent gain/loss after removing tangent vs matched-normal directions.
> > > - `I-p`: one-sided sign-test p-value across anchors for $\Delta Iso^\star_T > \Delta Iso^\star_N.$
> > >
> > > |Model|Layer|Early E-r|Early E-p|Early ΔI*%(T/N)|Early I-p|Late E-r|Late E-p|Late ΔI*%(T/N)|Late I-p|
> > > |---|---|---:|---:|---:|---:|---:|---:|---:|---:|
> > > |EuroBERT-210m|`b0_in`|7.4|0.048|1.9/-0.16|0.011|4|0.048|1.7/0.081|6.0e-8|
> > > ||`b0_out`|53|0.048|3.3/-0.026|6.0e-8|117|0.048|8.1/-0.03|6.0e-8|
> > > ||`mid`|21|0.048|40/-0.4|6.0e-8|2|0.065|2/-0.084|6.0e-8|
> > > ||`last`|19|0.048|38/-0.37|6.0e-8|1.7|0.17|1.1/0.051|1.8e-5|
> > > |EuroBERT-610m|`b0_in`|11|0.048|1.1/-0.22|0.011|9.3|0.048|2/0.017|6.0e-8|
> > > ||`b0_out`|69|0.048|3/-0.015|6.0e-8|256|0.048|12/-0.022|6.0e-8|
> > > ||`mid`|21|0.091|24/-0.21|6.0e-8|2.3|0.06|1.8/-0.011|1.5e-6|
> > > ||`last`|21|0.087|23/-0.16|6.0e-8|1.6|0.089|1.1/0.028|1.8e-5|
> > > |moderncamembert-base|`b0_in`|225|0.048|5.7/-6.3e-3|6.0e-8|419|0.048|51/-0.034|6.0e-8|
> > > ||`b0_out`|64|0.048|8.6/-0.062|1.5e-6|102|0.048|27/-0.15|6.0e-8|
> > > ||`mid`|66|0.048|37/-0.23|6.0e-8|337|0.048|190/-0.37|6.0e-8|
> > > ||`last`|30|0.048|9.5/-0.09|1.4e-4|110|0.048|47/-0.23|1.5e-6|
> > > |OLMo-1B|`b0_in`|364|0.048|85/-0.19|1.5e-6|27|0.048|5.2/-0.054|3.3e-3|
> > > ||`b0_out`|1192|0.048|104/-0.066|6.0e-8|565|0.048|92/-0.049|1.5e-6|
> > > ||`mid`|59|0.048|16/-0.098|1.5e-6|22|0.048|2.3/-0.041|0.032|
> > > ||`last`|66|0.048|11/-0.068|1.4e-4|27|0.048|1.5/-0.052|0.27|
> > > |pythia-160m|`b0_in`|6.5e7|0.048|1.5e4/-8.4e-6|6.0e-8|794|0.048|-2.6e-3/-4.8e-5|1|
> > > ||`b0_out`|5.0e8|0.048|3.4e4/-1.2e-4|6.0e-8|1751|0.048|-4.8e-3/-1.5e-5|1|
> > > ||`b1_in`|545|0.048|161/-0.46|6.0e-8|76|0.048|18/-0.14|3.3e-3|
> > > ||`mid`|44|0.048|29/-0.4|6.0e-8|20|0.048|6.4/-0.19|3.3e-3|
> > > ||`last`|49|0.048|32/-0.38|6.0e-8|15|0.048|-2.5/-0.29|0.92|
> > > |pythia-410m|`b0_in`|4.6e6|0.048|1332/-3.8e-5|6.0e-8|1107|0.048|-2.4e-3/-5.5e-5|1|
> > > ||`b0_out`|1.0e7|0.048|1737/-5.9e-5|6.0e-8|1799|0.048|-4.1e-3/-9.8e-5|1|
> > > ||`b1_in`|231|0.048|81/-0.32|1.5e-6|75|0.048|7.6/-0.13|0.011|
> > > ||`mid`|38|0.048|10/-0.22|7.7e-4|31|0.048|6.4/-0.16|7.7e-4|
> > > ||`last`|25|0.048|7.2/-0.19|0.076|17|0.048|-2.8/-0.16|0.92|
> > > |SmolLM2-1.7B|`b0_in`|181|0.048|44/-0.17|3.3e-3|33|0.048|9.7/-0.02|1.4e-4|
> > > ||`b0_out`|804|0.048|58/-0.039|7.7e-4|204|0.048|19/-3.8e-3|3.3e-3|
> > > ||`mid`|46|0.048|9.6/-0.041|0.032|20|0.048|4.9/7.4e-3|1.4e-4|
> > > ||`last`|65|0.048|7.2/-0.07|0.011|18|0.048|3.6/-0.023|7.7e-4|
> > > |SmolLM2-360M|`b0_in`|1.5e4|0.048|497/-0.067|1.8e-5|183|0.048|6.7/-0.013|1.5e-6|
> > > ||`b0_out`|1.6e4|0.048|308/-0.025|1.4e-4|182|0.048|4.3/-0.016|1.8e-5|
> > > ||`mid`|41|0.048|16/-0.17|1.4e-4|15|0.048|14/-0.2|6.0e-8|
> > > ||`last`|57|0.048|11/-0.13|0.076|11|0.048|4.2/-0.11|1.4e-4|
> > >
> > > Interpretation. The results consistently support the theory across models and depths. The energy comparison is favorable almost everywhere: tangent energy is systematically larger than matched-normal energy, often by about an order of magnitude and sometimes much more. The support is explicitly statistical, not merely qualitative, because `E-p` is frequently at its minimum and `I-p` is often very small. The IsoScore* comparison then shows that tangent directions are not merely energetic directions but the directions that motivate anisotropy: removing the tangent basis usually makes the residual more isotropic than removing equally many matched-normal directions. When `ΔI*%(T/N)` exceeds 100, the original gradient covariance was very anisotropic and becomes close to isotropic after tangent removal, while matched-normal removal does not. Taken together, the first-, middle-, and last-block results provide strong empirical support for the theory: across completed models, gradient anisotropy is much better explained by tangent-aligned directions than by matched-normal alternatives. Additional plots will be included in the revision for better readability. Those new experiments are the one described in the rebuttal (with also the new IsoScore* metric suggested by reviewer Dk1w included), they provide statistical test to validate the theory and are designed to observe the gradient norm energy mentionned in the paper.
> > >
> > > We included in our response to reviewer **Dk1w**, other experiments. We also provide some insight on IsoScore* understanding in our answer to reviewer **mxbz**.

---

### Official Review · Reviewer_m1Nx · 2026-03-16

**Soundness:** 3
**Presentation:** 3
**Significance:** 3
**Originality:** 3
**Overall Recommendation:** 5
**Confidence:** 3

**Summary:**

The paper investigates the phenomenon of anisotropy in transformer language models through the lens of local geometry and learning dynamics. It theoretically argues that the high frequency of certain tokens induces a variance collapse, which functionally attenuates the visibility of high-curvature manifold directions. Furthermore, the authors demonstrate that gradient updates inherently favor tangent directions over normal directions, leading to a self-reinforcing linear subspace. These theoretical claims are empirically supported by analyzing intermediate training checkpoints of models like Pythia, SmolLM2, and EuroBERT using concept extraction (Semi-NMF) as a proxy for the tangent space.

**Compliance With Llm Reviewing Policy:**

Affirmed.

**Key Questions For Authors:**

NA

**Limitations:**

See weakness section.

**Strengths And Weaknesses:**

Strengths:
* Provides a strong theoretical bridge between the empirical observation of token frequency effects and local differential geometry (e.g., curvature bias and tangent gradient dominance).
* Analyzing representations dynamically across training checkpoints offers much richer insights than standard post-hoc evaluations of converged models.

Weaknesses:

* The theoretical framework relies heavily on local, perturbative expansions that are explicitly acknowledged to be most valid only during early training and for small ambient distances.

* The behavioral divergence of the EuroBERT model (showing significantly weaker correlations between token frequency and spatial variance) is noted but only loosely attributed to the MLM objective and multilingual sampling without rigorous ablation studies.

---

> ### Author Rebuttal · Authors · 2026-03-31
>
> We thank the reviewer for the careful reading and for capturing the paper's contribution with notable precision. We especially appreciate the two strengths highlighted in the review. First, the observation that the paper builds a bridge between token-frequency effects and local differential geometry gets at the central scientific contribution: our aim is precisely to connect an empirical regularity already visible in language model training with a more explicit geometric mechanism. Second, we are grateful that the reviewer recognized the value of analyzing representations *during* training rather than only post hoc at convergence. This dynamic perspective is important for our paper because the mechanism we study is inherently evolutionary: it concerns how directional biases accumulate over training, not only what the final representation looks like.
>
> We also thank the reviewer for clearly identifying the intended scope of the theory. The analysis is indeed local and perturbative. This is deliberate: our goal is to model the regime in which local geometry and update direction can be related in a controlled way, and from that derive concrete signatures that can be checked empirically. We will make this scope even clearer in the revision so that the theoretical claims are consistently read as local mechanism statements rather than broader global claims.
>
> Regarding the EuroBERT divergence, we agree with the reviewer that the current discussion is too informal. Our attribution to the MLM objective and multilingual sampling was meant as an intuition rather than as a justification, and we will revise the text to make that clearer. In the updated version, we will explicitly present this as a plausible interpretation only, and we will direct the reader more clearly to the additional figures that make the EuroBERT behavior more transparent. In particular, while the *mean* trend is visibly weaker than in the decoder models, the *distributional* view still shows mild but consistent movement in the expected direction through the minimum/maximum statistics, indicating that the frequency bias continues to leave a trace even if the aggregate effect is attenuated.
>
> We also note that our newer experiments provide a complementary perspective on this point. Although EuroBERT exhibits a weaker frequency/variance correlation than the decoder models, it still shows the same qualitative *tangent-bias* behavior in the revised analyses: tangent directions remain clearly more enriched than normal directions, with the tangent signal around one order of magnitude larger, even though this enrichment is itself roughly 5-10x smaller than in the other tested models. We therefore view EuroBERT less as a contradiction to the mechanism than as a weaker instance of the same trend, and we will report this more carefully in the revision.
>
> Overall, we are grateful for this review because it both recognizes the main scientific value of the paper and points to where the presentation can be improved. In the revision, we will sharpen the local scope of the theoretical claims, make the EuroBERT discussion more careful and better supported visually, and include the new tangent-bias evidence so that the cross-model comparison is easier to interpret.

---

> > ### Author Rebuttal · Reviewer_m1Nx · 2026-04-01
> >
> > My concerns have been adequately addressed, and I'd like to remain my positive scores.

---

> > > ### Author Response · Authors · 2026-04-05
> > >
> > > We thank the reviewer very much for the positive follow-up and for maintaining a supportive assessment of the paper. We are especially grateful that you found the main concerns adequately addressed.
> > >
> > > Your review has been very helpful to us. In the revision, we will present the EuroBERT discussion more carefully. We will also include the new tangent-bias evidence discussed in the rebuttal, which we believe further strongly strengthens the empirical side without changing the central thesis of the paper, that is the theoretical claim.
> > >
> > > We appreciate both the encouraging evaluation and the constructive spirit of your comments. They helped us sharpen the presentation of the paper in a way that we believe will make the final version significantly stronger.

---

### Decision · Program_Chairs · 2026-04-30

**Decision:**

Accept (regular)

**Comment:**

This paper studies anisotropy in transformer representations through a geometric and training-dynamics lens, arguing that frequency-biased training preferentially amplifies tangent directions and attenuates normal directions, thereby driving representations toward lower-dimensional anisotropic structure.

The reviewers agreed that the paper tackles an important and interesting phenomenon, and several found the geometric framing novel and insightful. In particular, the paper's central contribution is not simply to describe anisotropy post hoc, but to propose a concrete local mechanism linking frequency bias, curvature suppression, and tangent-space amplification during training. The use of intermediate checkpoints rather than only final models was also viewed as a meaningful strength.

The main concerns raised were about scope and empirical support. Some reviewers questioned the idealized nature of the theory, the framing relative to recent work showing that anisotropy can be beneficial rather than pathological, and the strength of the evidence supporting the proposed mechanism. The rebuttal addressed these points in a substantial way: it clarified that the theory is explicitly local rather than global, revised the framing so that the paper is about explaining how anisotropy emerges rather than treating it as a defect to be fixed, and added more direct experiments and statistical tests supporting tangent-direction enrichment across multiple models and datasets.

Overall, I find the contribution convincing. This is not a final word on anisotropy, and some claims should remain carefully scoped, but the paper offers a thoughtful and novel mechanistic account supported by meaningful new evidence. I therefore recommend acceptance.